# In-situ low-temperature sulfur CVD on metal sulfides with $SO_2$ to realize self-sustained adsorption of mercury

Qinyuan Hong [1,4], Haomiao Xu [1,4] ✉, Xiaoming Sun[1], Jiaxing Li[1], Wenjun Huang[1], Zan Qu [1,2] ✉, Lizhi Zhang [1,3] & Naiqiang Yan [1,2] ✉

Capturing gaseous mercury ($Hg^0$) from sulfur dioxide ($SO_2$)-containing flue gases remains a common yet persistently challenge. Here we introduce a low-temperature sulfur chemical vapor deposition (S-CVD) technique that effectively converts $SO_2$, with intermittently introduced $H_2S$, into deposited sulfur ($S_d^0$) on metal sulfides (MS), facilitating self-sustained adsorption of $Hg^0$. ZnS, as a representative MS model, undergoes a decrease in the coordination number of Zn–S from 3.9 to 3.5 after $S_d^0$ deposition, accompanied by the generation of unsaturated-coordinated polysulfide species ($S_n^{2-}$, named $S_d^*$) with significantly enhanced $Hg^0$ adsorption performance. Surprisingly, the adsorption product, HgS (ZnS@HgS), can serve as a fresh interface for the activation of $S_d^0$ to $S_d^*$ through the S-CVD method, thereby achieving a self-sustained $Hg^0$ adsorption capacity exceeding 300 mg g$^{-1}$ without saturation limitations. Theoretical calculations substantiate the self-sustained adsorption mechanism that $S_8$ ring on both ZnS and ZnS@HgS can be activated to chemical bond $S_4$ chain, exhibiting a stronger $Hg^0$ adsorption energy than pristine ones. Importantly, this S-CVD strategy is applicable to the in-situ activation of synthetic or natural MS containing chalcophile metal elements for $Hg^0$ removal and also holds potential applications for various purposes requiring MS adsorbents.

Mercury pollution has aroused global concern mainly ascribed to the volatility, insolubility, and long-range transport of gaseous elemental mercury ($Hg^0$), which is the focus of mercury abatement[1–4]. Effective control of $Hg^0$ pollutant necessitates the use of functional materials that exhibit high activity, stability, and tolerance to flue gas conditions[5–7]. Recent developments in materials for $Hg^0$ capture include carbon-based[8,9], oxide-based[10], and noble metal-based materials[11,12]. However, the presence of $SO_2$ in any flue gas has been known to negatively impact $Hg^0$ removal, leading to surface sulfation or the occupation of active sites[13]. More unfortunately, the non-ferrous

metal smelting industry is considered as the largest single source of $Hg^0$ emissions, where high concentrations of $SO_2$ and $Hg^0$ co-exist[14]. Therefore, achieving large-capacity adsorption of $Hg^0$ at high concentration of $SO_2$ remains a significant challenge.

While many sulfur-based materials have shown a degree of resistance to $SO_2$ in $Hg^0$ adsorption[15,16], their capacities often experience significant suppression due to active site depletion or deactivation, particularly at high $SO_2$ concentrations[17]. This limitation frequently requires off-line regeneration under harsh conditions (e.g., heating or acidification treatments with irreversible destruction) or even

[1]School of Environmental Science and Engineering, Shanghai Jiao Tong University, Shanghai 200240, China. [2]Shanghai Institute of Pollution Control and Ecological Security, Shanghai 200092, China. [3]Key Laboratory of Pesticide & Chemical Biology of Ministry of Education Institute of Applied & Environmental Chemistry College of Chemistry, Central China Normal University, Wuhan 430079, China. [4]These authors contributed equally: Qinyuan Hong, Haomiao Xu. ✉e-mail: xuhaomiao@sjtu.edu.cn; quzan@sjtu.edu.cn; nqyan@sjtu.edu.cn

necessitates the replacement of adsorbents[18,19]. To address the challenge of active depletion or deactivation, a more cost-effective and convenient method involves the continuous replenishment of active sites in situ at the interface of sulfur-based materials. Recognizing that the performance of metal sulfides (MS) relies heavily on the quantity of active sulfur sites[20], a viable strategy is to directly convert $SO_2$ to sustainably replenish surface active sites, thereby turning the negative effect of $SO_2$ into a positive one. However, the high average S−O bond energy of $SO_2$ (548 kJ mol⁻¹) necessitates high-temperature conditions (>2000 °C) for its decomposition[21,22], whereas the preferential reaction of $SO_2$ with flue gas $O_2$ impedes the feasibility of this pathway[23]. Notably, the assistance of $H_2S$ can lower the S−O bond breaking energy barrier (139 kJ mol⁻¹) and further reorganize the S−S bond to generate elemental sulfur through the Claus reaction[24,25]. Fortunately, $H_2S$ or its raw materials ($Na_2S$ or $NaHS$) are easily accessible and commonly used for heavy metals removal from various wastewaters in non-ferrous smelters[26,27].

However, two key challenges persist in achieving our objectives. Firstly, it involves effectively generating fresh sulfur on MS surface. More importantly, it pertains to activating the deposited sulfur ($S_d^0$) for site replenishment instead of allowing it to aggregate into the inert $S_8^0$ state (octatomic ring structure with poor $Hg^0$ adsorption activity[28,29]). To address these challenges, we have developed a sulfur chemical vapor deposition (S-CVD) method using the Claus reaction between excessive $SO_2$ in flue gases and intermittently added $H_2S$. This method facilitates the deposition of gas-phase sulfur species with high controllability and scalability. Furthermore, an active interface is crucial to bonding with $S_d^0$ to create unsaturated coordination sites rather than coordination-saturated $S_8$. Notably, the incorporation of anchoring sites to bond with sulfur can maintain its unsaturated state[30]. Chalcophile elements exhibit a natural tendency to lose outer electrons to form an 18-electron outermost structure ($s^2p^6d^{10}$), which

in turn combines with sulfur ($3s^23p^4$) to form an ionic compound under ambient conditons[31,32]. Thus, MS containing chalcophile metals emerges as promising candidates to bond with $S_d^0$, preventing it from falling into a saturated-coordinated ring structure.

Hence, this work employs the proposed in-situ S-CVD method on chalcophile MS to counteract the negative effects of $SO_2$ and achieve self-sustained $Hg^0$ adsorption, enabling in-situ reactivation without the need to replace spent adsorbents. Variety of experimental conditions and characterization methods, such as scanning electron microscopy (SEM), X-ray absorption fine structure (XAFS), and density functional theory (DFT) calculations, were devoted to evaluating the self-sustained adsorption performance, revealing the deposition process of $S_d^0$, identifying the formation of unsaturated coordination environments, and calculating the energy changes to interpret the self-sustained adsorption mechanism. The results indicate that $S_d^0$ can be efficiently activated to polysulfide ($S_n^{2-}$, named $S_d^*$) species by chalcophile MS, including the formed HgS itself, ensuring self-sustained $Hg^0$ adsorption. This in-situ S-CVD approach provides a promising solution to active site depletion and poisoning issues and offers an avenue for efficient and continuous heavy metal removal using MS materials.

## Results

### Establishment of in-situ S-CVD method for $SO_2$ deposition

The in-situ S-CVD method was established for flue gas $SO_2$ deposition. To initiate S-CVD, a small amount of $H_2S$ (100 ppm) was injected into the $SO_2$-containing flue gas upstream of $Al_2O_3$@MS adsorbents (Supplementary Fig. 1). In actual non-ferrous smelting processes, flue gas particle-bond mercury ($Hg^p$) and oxidized mercury ($Hg^{2+}$) can be respectively removed by an electrical precipitator and scrubber, resulting in a subsequent flue gas with high concentrations of $SO_2$ and $Hg^0$ (Supplementary Fig. 2). Extraction of approximately 0.1‰ of total

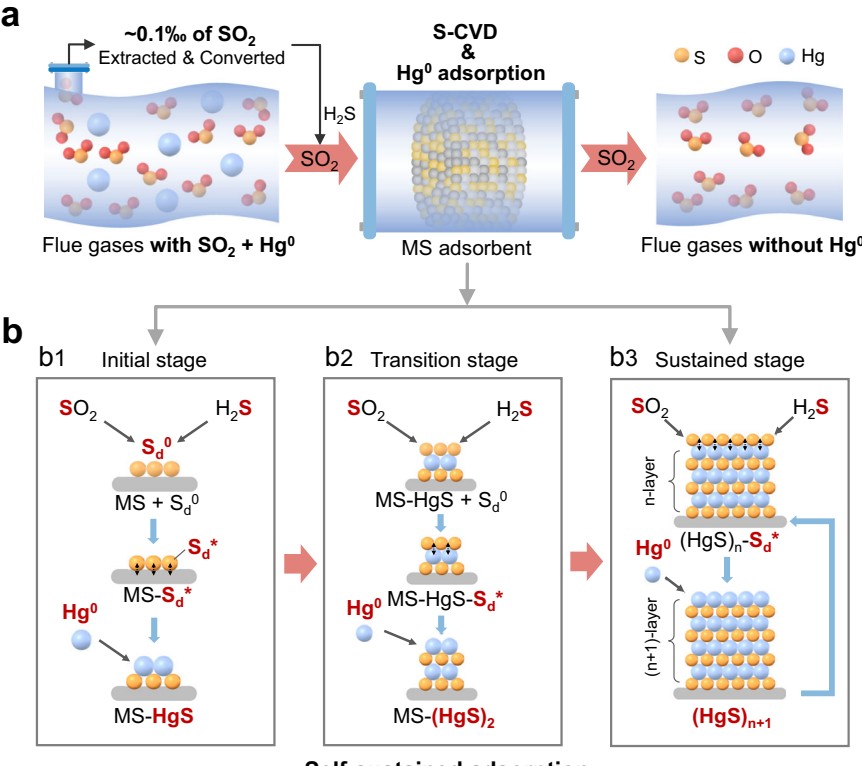

**Fig. 1 | Proposed in-situ S-CVD strategy for $Hg^0$ self-sustained adsorption on metal sulfides. a** $Hg^0$ removal through proposed in-situ S-CVD strategy in smelting flue gas. **b** Schematic illustration of the $Hg^0$ self-sustained adsorption on metal sulfides. **b1** Initial stage, $S_d^0$ activated only by MS; **b2** Transition stage, $S_d^0$ activated by MS and/or HgS; **b3** Sustained stage, $S_d^0$ activated by HgS itself.

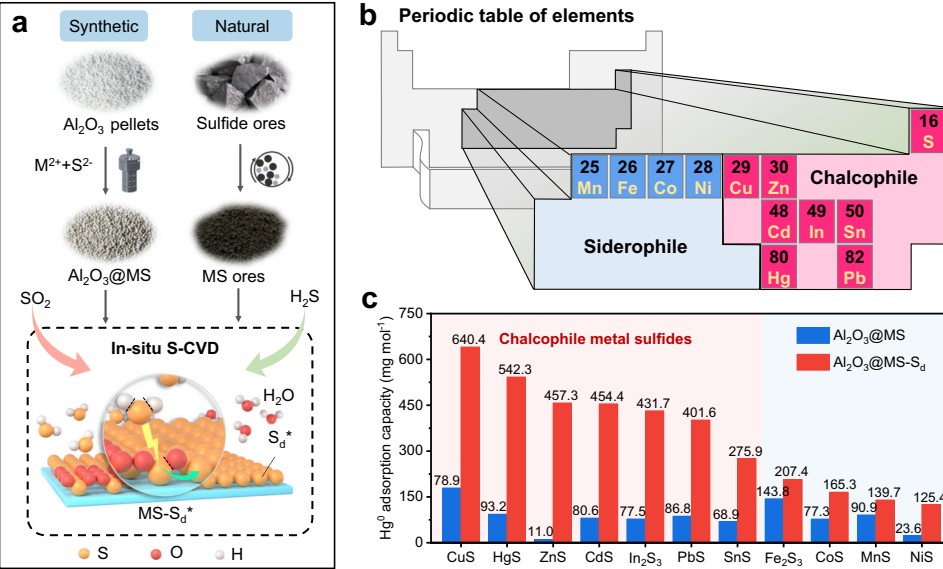

**Fig. 2 | Enhanced performance of different metal sulfides using in-situ SCVD activation. a** Proposed in-situ S-CVD strategy on synthesized metal sulfides or natural sulfide ores. **b** Geochemical classification of elements containing chalcophile and siderophile elements. **c** $Hg^0$ adsorption capacities of different $Al_2O_3$@MS and $Al_2O_3$@MS-$S_d$ (M = Cu, Hg, Zn, Cd, In, Pb, Sn, Ni, Co, Fe, and Mn). Reaction conditions: adsorbent mass = 0.3 g, total flow rate = 360 mL min$^{-1}$, $SO_2$ concentration = 5000 ppm (during S-CVD process), $H_2S$ concentration = 100 ppm (during S-CVD process), S-CVD time = 15 min, $Hg^0$ concentration = $(1.5 \pm 0.05)$ mg m$^{-3}$, reaction temperature = 80 °C, and reaction time = 180 min.

$SO_2$ for on-site conversion to $H_2S$[27,33] can satisfy the needs of S-CVD, and $Hg^0$ will be removed by the proposed self-sustained adsorption method on adsorbents (Fig. 1a). In this method, MS functions as the fresh surface for the initial S-CVD and $Hg^0$ adsorption, and then the spent MS (i.e., MS-HgS) acts as new surface for further S-CVD and $Hg^0$ adsorption, ultimately achieving the sustained adsorption by HgS itself (Fig. 1b).

During the S-CVD process, the reaction ratio of $H_2S$ and $SO_2$ was monitored as 2.1: 1 and chemical composition analysis exhibited that the sulfur content in different $Al_2O_3$@MS increased by 2.1%–3.0% after 180 min of S-CVD (Supplementary Fig. 3). These confirm the occurrence of the Claus reaction on the $Al_2O_3$@MS surface, which is a critical step in the S-CVD process. Furthermore, the optimal adding sequence of $H_2S$ and $SO_2$ was investigated to understand the generation mechanism of $S_d^0$. The results demonstrated that the activity of $Al_2O_3$@MS pretreated with $H_2S$ followed by $SO_2$ was significantly lower than that pretreated with $SO_2$ followed by $H_2S$ (Supplementary Fig. 4). Meanwhile, pretreatment only by $SO_2$ cannot enhance the activity of $Al_2O_3$@MS. This finding indicates that the formation of $S_d^0$ on $Al_2O_3$@MS surface followed the Eley-Rideal mechanism, in which $SO_2$ is first adsorbed on adsorbent surface and then reacts with gaseous $H_2S$ to produce $S_d^0$ (Eq. 1, 2):

$$SO_2(g) + Al_2O_3@MS \rightarrow Al_2O_3@MS\text{-}SO_2(ads) \tag{1}$$

$$Al_2O_3@MS\text{-}SO_2(ads) + 2H_2S(g) \rightarrow Al_2O_3@MS\text{-}3S_d^0 + 2H_2O \tag{2}$$

Importantly, the negative Gibbs free energy ($\Delta G^0 = -91$ kJ mol$^{-1}$ at 25 °C) of the Claus reaction and the much lower concentration of $H_2S$ (100 ppm) compared to $SO_2$ ($\geq$5000 ppm) used in S-CVD guarantee the sufficient reaction of added $H_2S$.

## Deposited sulfur activation on MS for $Hg^0$ removal

The surface properties of metal sulfides play a vital role in S-CVD process. Both synthetic and natural MS served as the deposition surface (Fig. 2a). In light of Goldschmidt geochemical classification of the

elements, metals with chalcophile nature have higher affinity towards sulfur, which could accelerate the stimulation of $S_d^0$ (refs. 31,32). Thus, a range of typical chalcophile metal elements, including Cu, Zn, In, Cd, Pb, and Sn, were chosen as candidates for the synthesis of MS, while some siderophile metals, including Mn, Fe, Co, and Ni, were offered as contrasts (Fig. 2b). As depicted in Fig. 2c and Supplementary Table 1, after 15 min of S-CVD, various $Al_2O_3$@MS-$S_d$ showed substantial differences in enhancing their $Hg^0$ adsorption performances, of which all the chalcophile $Al_2O_3$@MS-$S_d$ exhibited significantly increase in their adsorption capacities. Notably, $Al_2O_3$@CuS-$S_d$ demonstrated a remarkable increase the $Hg^0$ adsorption capacity, rising from 178.9 to 640.4 mg mol$^{-1}$(within 180 min and normalized to MS molar mass). Similarly, $Al_2O_3$@ZnS-$S_d$ exhibited a significant enhancement, with adsorption capacity increasing from 11.0 to 457.3 mg mol$^{-1}$. $Al_2O_3$@CdS-$S_d$, $Al_2O_3$@In$_2$S$_3$-$S_d$, $Al_2O_3$@PbS-$S_d$, and $Al_2O_3$@SnS-$S_d$ also showed considerable improvements in their $Hg^0$ adsorption capacities, reaching 454.4, 431.7, 401.6, and 275.9 mg mol$^{-1}$, respectively. However, the adsorption capacities of $Al_2O_3$@Fe$_2$S$_3$-$S_d$, $Al_2O_3$@CoS-$S_d$, $Al_2O_3$@MnS-$S_d$, and $Al_2O_3$@NiS-$S_d$ were comparatively lower, reaching 207.4, 165.3, 139.7, and 125.4 mg mol$^{-1}$, respectively. In addition, the $Al_2O_3$-$S_d$, $Al_2O_3$@ZnSO$_4$-$S_d$, and $Al_2O_3$@Na$_2$S-$S_d$ did not show improved performance compared to their raw materials (Supplementary Fig. 5), implying that the isolated presence of metal or sulfur sites cannot directly activate the deposited $S_d^0$.

Further, to quantitatively explore the role of metal sites in MS on the activation of $S_d^0$, we further construct the relationship between the $Hg^0$ adsorption capacity increment ($Q_i = Q_{Al2O3@MS\text{-}Sd} - Q_{Al2O3@MS}$) and metal-sulfur (M–S) bond energy (measured by bond length[34], Supplementary Table 2) of the investigated metal sulfides. As depicted in Supplementary Fig. 6, the $Q_i$ of chalcophile MS showed a negative relationship with the increase of M–S bond length. However, for siderophile MS, higher M–S bond length instead led to relatively higher $Q_i$. It can be deduced that the activity of $S_d^0$ on the sulfide interface highly depends on the geochemical characteristics of metal element and the M–S affinity. Notably, Hg also belongs to chalcophile elements (Fig. 2b), and it is supposed that HgS itself could potentially play a role in activating $S_d^0$ (see later in Fig. 5e).

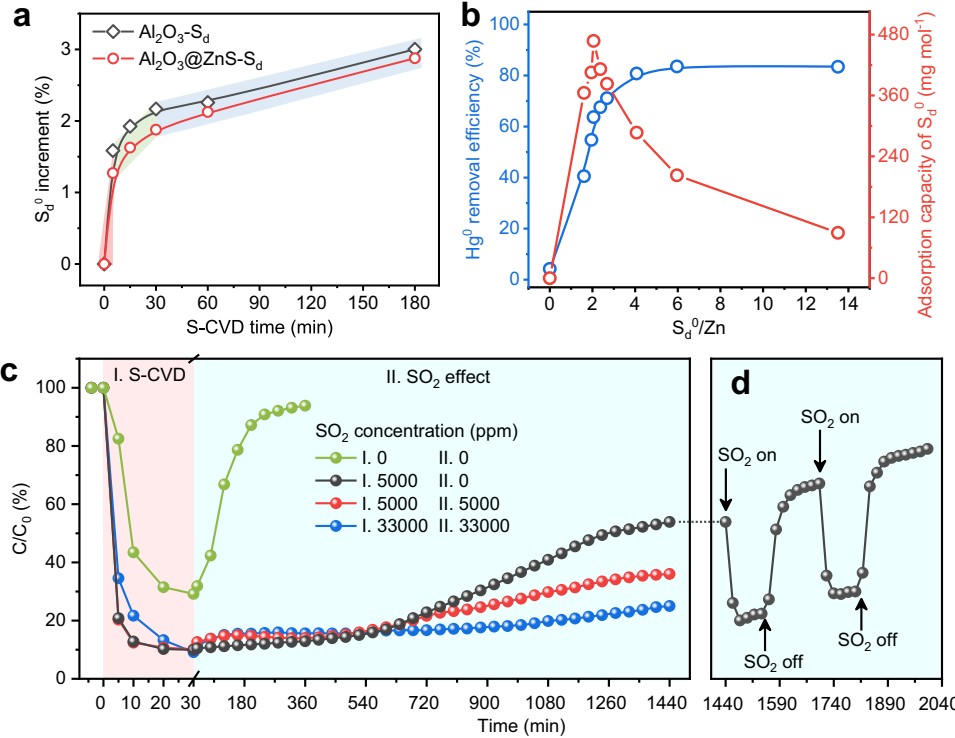

**Fig. 3 | Influencing factors of in-situ SCVD strategy for Hg⁰ adsorption over Al₂O₃@ZnS-S_d.** **a** $S_d^0$ increment in $Al_2O_3$ and $Al_2O_3@ZnS$ with the increase of S-CVD time. The red, green and blue-gray highlights represent three stages of different $S_d^0$ growth rates. **b** The relationship between $S_d^0$ increment and Hg⁰ adsorption capacity of $Al_2O_3@ZnS$-$S_d$. **c** The effect of SO₂ concentration on Hg⁰ adsorption breakthrough curve of $Al_2O_3@ZnS$-$S_d$. **d** The effect of intermittent addition of SO₂ on the Hg⁰ adsorption curve over $Al_2O_3@ZnS$-$S_d$. Reaction conditions: adsorbent mass = 0.3 g, total flow rate = 360 mL min⁻¹, Hg⁰ concentration = (1.5 ± 0.05) mg m⁻³, SO₂ concentration = 5000 ppm (for **a** and **b**), H₂S concentration = 100 ppm (during the S-CVD process), temperature = 80 °C (for **a**), and S-CVD time = 30 min (for **b** and **c**).

We then choose ZnS as a representative model to optimize the reaction conditions owing to its significant performance enhancement (~42 times) and simple metal and sulfur speciation. The sulfur content in $Al_2O_3@ZnS$-$S_d$ exhibited a three-stage growth pattern with increasing S-CVD time (Fig. 3a). The growth rate of $S_d^0$ decreased from 2.5 mg g⁻¹ min⁻¹ at 0–5 min to 0.2 mg g⁻¹ min⁻¹ at 5–30 min then to 0.06 mg g⁻¹ min⁻¹ at >30 min (Supplementary Table 3). The decreasing formation rate indicates that the fresh surface of the adsorbent was gradually covered by $S_d^0$, and once the surface was completely covered, subsequent $S_d^0$ would generate on the existing $S_d^0$ layer, resulting in a final slow but steady growth rate. Figure 3b presents the relationship between $S_d^0$ increment and Hg⁰ removal efficiency of $Al_2O_3@ZnS$-$S_d$. The results showed that the removal efficiency gradually increased to 71.7% when the ratio of $S_d^0$/Zn increased to 2.7 (corresponding to 60 min of S-CVD); while, as the ratio further increased to 4.1 (240 min), 6.0 (480 min), and 13.5 (1440 min), the removal efficiency remained at a stable level of around 83%. This indicates that excess deposited $S_d^0$ did not work, presumably attributed to the inevitable aggregation of excess $S_d^0$ into inert $S_8^0$ (ref. 29). When the adsorption capacity is normalized to the mole of $S_d^0$ (red curve in Fig. 3b), it reached a maximum of 467.5 mg mol⁻¹ at the $S_d^0$/Zn ratio of 2.1 (15 min) and then gradually decreased to 89.3 mg mol⁻¹ as the ratio increased to 13.5. Therefore, 30 min of S-CVD was chosen as the optimal condition by integrating the Hg⁰ removal efficiency and $S_d^0$ utilization. Besides S-CVD time, reaction temperatures and other flue gas components also played significant roles. Thermogravimetric analysis (TGA) results indicated the high thermal stability of $Al_2O_3@ZnS$-$S_d$ below 200 °C (Supplementary Fig. 7). The increase in adsorption temperature from 60 to 120 °C improved the Hg⁰ removal efficiency of $Al_2O_3@ZnS$-$S_d$ from 23.8% to 89.9% within 180 min (Supplementary Fig. 8). However, further elevation of temperature to 140 °C and 160 °C

resulted in decreasing activity to 86.6% and 75.9%, respectively, presumably due to the re-decomposition of partially captured Hg⁰ (Supplementary Fig. 9). Moreover, $Al_2O_3@ZnS$-$S_d$ demonstrated high tolerance to different gas components (Supplementary Fig. 10). The addition of 5000 ppm SO₂ in adsorption process resulted in enhanced removal efficiency by 3.2% at 120 °C, and further addition of 5% O₂ or 100 ppm NO had slight influence with a reduction of 2.6% and 6.8%, respectively. The introduction of 5000 ppm SO₂ + 4% H₂O showed negative effect on Hg⁰ adsorption performance of $Al_2O_3@ZnS$-$S_d$, presumably due to the competing adsorption between H₂O and Hg⁰ on the active sites[35] and the hydrophilicity of $Al_2O_3$ (ref. 36); however, its removal efficiency maintained at a stable level of ~70% without reduction for 180 min adsorption.

Long-term experiments were conducted to clarify the SO₂ effect on Hg⁰ removal during the S-CVD and adsorption process under the optimal condition (H₂S = 100 ppm, temperature = 120 °C, S-CVD time = 30 min) (Fig. 3c). In stage I (S-CVD process), without the presence of SO₂, $Al_2O_3@ZnS$ initially exhibited a temporary increase in Hg⁰ removal but rapidly decreased once the S-CVD was stopped. This improvement might be attributed to the sulfuration of residual metal salt precursors on the surface. However, when SO₂ was added during the S-CVD process, the Hg⁰ removal sharply increased to around 90%. Furthermore, SO₂ continued to exhibit an extraordinary effect on the performance of $Al_2O_3@ZnS$-$S_d$ after the S-CVD process. As shown in stage II (without H₂S), the slope of curve without SO₂ addition in the adsorption process (black line) was lower than that with SO₂ addition (red one), indicating a positive effect of SO₂, which is contrary to our traditional perceptions. Moreover, as the SO₂ concentration increased from 0 ppm to 5000 ppm and to 33,000 ppm, the Hg⁰ removal after 1440 min reaction increased from 46.1% to 64.0% and to 75.0%, respectively. To verify this unexpected

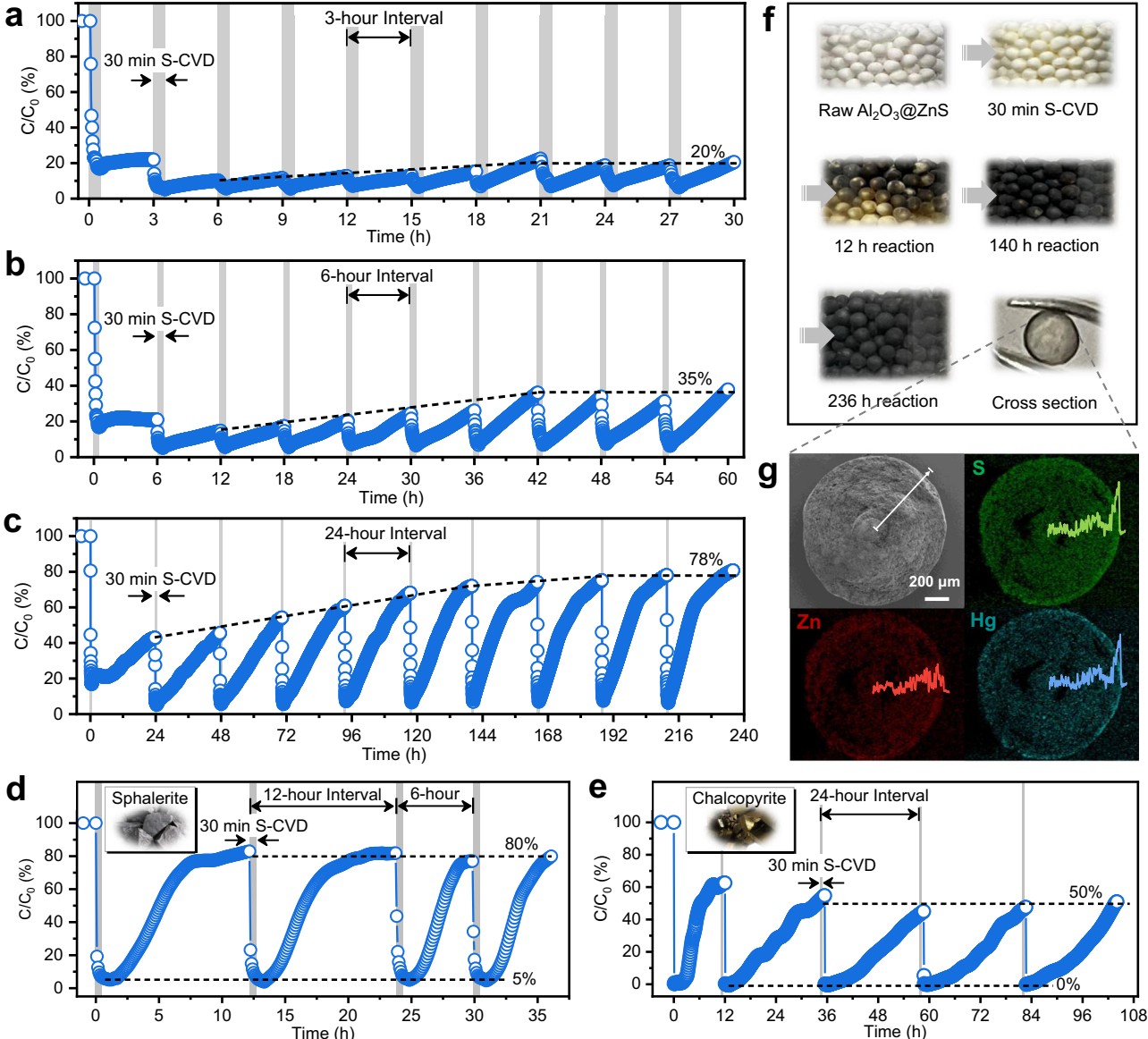

**Fig. 4 | Self-sustained adsorption performance of Hg⁰.** Hg⁰ adsorption break-through curves of $Al_2O_3$@ZnS-$S_d$ assisted with (**a**) 3 h, **b** 6 h, and **c** 24 h intermittent S-CVD. Hg⁰ adsorption breakthrough curves of (**d**) natural chalcopyrite ore and (**e**) sphalerite ore assisted with intermittent S-CVD. Insets in **d** and **e**: photographs of the corresponding natural ores. Reaction conditions: adsorbent mass = 0.4 g (for **a**–**c**) or 1 g (for **d** and **e**), temperature = 120 °C (for **a**–**d**) or 40 °C (for **e**),

[Hg⁰] = (2.5 ± 0.05) mg m⁻³, [$SO_2$] = 5000 ppm (for **a**–**c**) or 6% (for **d** and **e**), [$H_2O$] = 4%, [$H_2S$] = 100 ppm (during S-CVD), and total flow rate = 300 mL min⁻¹. **f** Photographs of $Al_2O_3$@ZnS in different reaction stages. **g** EDS mapping images of the cross-section of spent $Al_2O_3$@ZnS-$S_d$. Inset: line scanning results of the selected position.

positive effect, $SO_2$ was added intermittently to the $SO_2$-free system in stage II (Fig. 3d). The results showed that once 5000 ppm of $SO_2$ was introduced, the Hg⁰ removal efficiency was instantly improved by about 35%; however, once the $SO_2$ was turned off, it dropped back down to a lower level immediately. Thus, $SO_2$ was identified as the positive component for Hg⁰ adsorption over the $Al_2O_3$@ZnS-$S_d$. Through Hg⁰ temperature-programmed desorption (Hg⁰-TPD) experiments, we analyzed the changes in de-mercury products (Supplementary Fig. 9). The desorption peaks of spent $Al_2O_3$@ZnS-$S_d$ in the absence of $SO_2$ located at 210−265 °C, which are attributed to the decomposition temperature of β-HgS[37]. While, after adsorption in the presence of $SO_2$, there appeared a new peak centered at higher temperature of 365 °C, which is assigned to the decomposition temperature of α-HgS[38], thereby improving its Hg⁰ adsorption stability. Additionally, the performance of $Al_2O_3$@MS without S-CVD assistance was significantly reduced at high concentrations of

$SO_2$, excluding the promotional effect of $SO_2$ on $Al_2O_3$@MS itself (Supplementary Fig. 11).

## Self-sustained adsorption performance of $Al_2O_3$@MS-$S_d$

Although $Al_2O_3$@ZnS-$S_d$ exhibited high performance and $SO_2$ can further promote the Hg⁰ adsorption, it still has a limited number of active sites and requires replacement of the adsorbent once depleted. If we can periodically replenish the $S_d^0$ on the spent $Al_2O_3$@ZnS-$S_d$ surface, self-sustained adsorption of Hg⁰ can be achieved without the need for adsorbent replacement. Hence, simulated flue gas ((2.5 ± 0.05) mg m⁻³ Hg⁰, 5000 ppm $SO_2$, 4% $H_2O$, total flow rate = 300 mL min⁻¹, and reaction temperature = 120 °C) was applied to investigate the self-sustained adsorption performance of $Al_2O_3$@ZnS-$S_d$. 100 ppm of $H_2S$ was intermittently injected into the flue gas for 30 min per 3, 6, and 24 h to replenish $S_d^0$ (Fig. 4a–c). The Hg⁰ removal of $Al_2O_3$@ZnS-$S_d$ can reach around

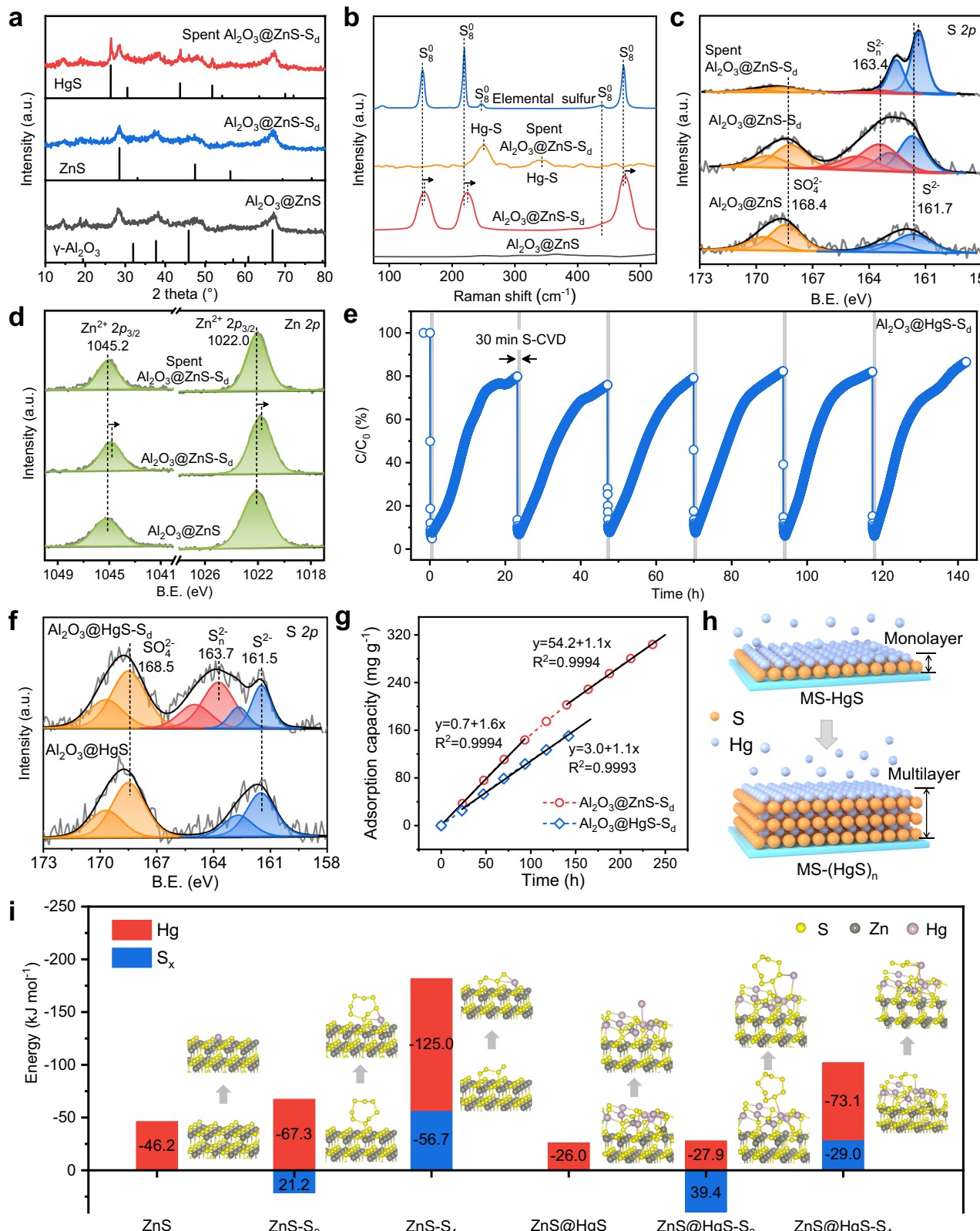

**Fig. 5 | Self-sustained adsorption mechanism. a** XRD patterns of Al$_2$O$_3$@ZnS, Al$_2$O$_3$@ZnS-S$_d$, and spent Al$_2$O$_3$@ZnS-S$_d$. **b** Raman spectra of Al$_2$O$_3$@ZnS, Al$_2$O$_3$@ZnS-S$_d$, spent Al$_2$O$_3$@ZnS-S$_d$, and pure S$_8$. **c** S 2$p$ and **d** Zn 2$p$ XPS spectra of Al$_2$O$_3$@ZnS, Al$_2$O$_3$@ZnS-S$_d$, and spent Al$_2$O$_3$@ZnS-S$_d$. **e** Hg$^0$ adsorption breakthrough curve of Al$_2$O$_3$@HgS-S$_d$ assisted with intermittent S-CVD. Reaction conditions: adsorbent mass = 0.4 g, [Hg$^0$] = (2.5 ± 0.05) mg m$^{-3}$, [SO$_2$] = 5000 ppm, [H$_2$O] = 4%, [H$_2$S] = 100 ppm (30 min per 24 h), and total flow rate = 300 mL min$^{-1}$. **f** S 2$p$ XPS spectra of Al$_2$O$_3$@HgS and Al$_2$O$_3$@HgS-S$_d$. **g** Linear fitting of the adsorption rates of Al$_2$O$_3$@ZnS-S$_d$ and Al$_2$O$_3$@HgS-S$_d$ at 120 °C. **h** Schematic illustration of in-situ intermittent S-CVD on Al$_2$O$_3$@MS surface for Hg$^0$ multilayer adsorption. **i** DFT calculations of S$_d^0$ activation and Hg$^0$ self-sustained adsorption behaviors on ZnS (111) surface.

95% after each round of S-CVD except the first round (83.2%). After the supplemented $S_d^0$ was consumed by $Hg^0$, the removal efficiency gradually decreased. Meanwhile, as the number of S-CVD increased, the 3, 6, and 24 h breakthrough ratios gradually converged to ~20%, ~35%, and ~78%, respectively. Additionally, owing to the higher performance of $Al_2O_3$@CuS-$S_d$ at low temperatures (Supplementary Fig. 12), it displayed the self-sustained adsorption performance at 60 °C, with an initial $Hg^0$ removal efficiency of ~90% and a 24 h breakthrough ratio of ~50% in each round of reaction (Supplementary Figs. 13). Moreover, $Al_2O_3$@ZnS-$S_d$ can restore its original activity after $Hg^0$ desorption and secondary S-CVD (Supplementary Fig. 14). Additionally, we evaluated the $Hg^0$ re-emission of $Al_2O_3$@ZnS-$S_d$ after ten rounds of reaction. At the reaction temperature, the $Hg^0$ re-emission concentration can be reduced from 1.7 mg m$^{-3}$ to 0.05 mg m$^{-3}$ (emission standard for non-ferrous smelting flue gas in China) in 30 min with the assistance of S-CVD, and once the temperature dropped to room temperature, the $Hg^0$ concentration rapidly decreased to 0 mg m$^{-3}$ (Supplementary Fig. 15). Thus, the S-CVD strategy not only can fulfill the self-sustained adsorption of $Hg^0$, but also inhibit the re-emission of adsorbed mercury.

Besides synthesized $Al_2O_3$@MS, natural sulfide ores that contain chalcophile metal elements, such as chalcopyrite and sphalerite, also enable their potential for self-sustained $Hg^0$ adsorption. As depicted in Fig. 4d, at conditions of near-actual flue gas $SO_2$ concentration (6%), natural sphalerite ore achieved ~95% $Hg^0$ removal after each round of S-CVD, and the breakthrough ratio converged to ~80%. Chalcopyrite exhibited enhanced $Hg^0$ adsorption performance with intermittent S-CVD at a lower temperature (40 °C), which reached a 100% initial $Hg^0$ removal efficiency and had a 24 h breakthrough ratio of ~50% in each round of reaction (Fig. 4e). Thus, directly utilization of natural sulfide ores as adsorbents can effectively lower the cost for $Hg^0$ pollution control as well as improve the adsorption capacity taking advantage of their self-sustained adsorption properties.

Figure 4f shows the macrophotographs of $Al_2O_3$@ZnS in different reaction stages. After the first round of S-CVD, the color of $Al_2O_3$@ZnS changed from white to pale yellow, verifying the formation of $S_d^0$. Then, the color of spent $Al_2O_3$@ZnS-$S_d$ gradually turned black after 140 h reaction (six rounds of S-CVD and $Hg^0$ adsorption). The cross-sectional view of spent $Al_2O_3$@ZnS-$S_d$ pellets (after ten rounds) in Fig. 4f depicted an obvious black shell. The energy dispersion X-ray spectroscopy (EDS) mapping images and the selected line scanning curves of the cross-section of $Al_2O_3$@ZnS revealed that the contents of S and Zn elements on $Al_2O_3$ pellet gradually decreased from outside in (Supplementary Fig. 16). After reaction, S and Hg elements were more scattered in the outer layer and exhibited synchronous linear increase at the boundary layer (~50 μm); while, there presented a decline in Zn element at this boundary layer (Fig. 4g). In addition, the S and Hg contents on the surface of the spent $Al_2O_3$@ZnS-$S_d$ was much higher than the Zn content (Supplementary Fig. 17). This indicates a layer-by-layer outward deposition of $S_d^0$ and adsorption of $Hg^0$ on $Al_2O_3$@ZnS surface. Moreover, assume that the Zn: S ratio in raw $Al_2O_3$@ZnS is 1:1, the ratio of Hg to $S_d^0$ in spent $Al_2O_3$@ZnS-$S_d$ was calculated as 1.04 in light of the ESD result of the cross-section (Supplementary Fig. 18), giving an indication that $S_d^0$ atoms were fully utilized for $Hg^0$ adsorption.

**Mechanism for self-sustained $Hg^0$ adsorption on $Al_2O_3$@MS-$S_d$**

Identifying the $S_d^0$ activation and $Hg^0$ adsorption behaviors on $Al_2O_3$@ZnS contributes to a profound insight into the self-sustained adsorption mechanism of $Hg^0$. The Brunauer−Emmett−Teller (BET) surface area, total pore volume, and average pore size of $Al_2O_3$, $Al_2O_3$@ZnS, and $Al_2O_3$@ZnS-$S_d$ were not significantly different (Supplementary Table 4), suggesting that the coating of ZnS and the deposition of $S_d^0$ did not affect the pore structure of

$Al_2O_3$. The X-ray diffraction (XRD) pattern of $Al_2O_3$@ZnS-$S_d$ showed that, besides the diffraction peaks assigned to γ-$Al_2O_3$ (JCPDS no. 79-1558) and sphalerite ZnS (JCPDS no. 77-2100), no peaks related to elemental sulfur emerged (Fig. 5a), suggesting the amorphous structure of formed $S_d^0$. The Raman spectrum of $Al_2O_3$@ZnS-$S_d$ charactered the peaks at 155.6, 223.9, 443.8, and 475.3 cm$^{-1}$, which are related higher than Raman shift of $S_8^0$ (Fig. 5b). This indicates that the surface ZnS can change its S−S vibration of $S_d^0$ and prevent its aggregation[39]. In the X-ray photoelectron spectroscopy (XPS) S 2$p$ spectrum of $Al_2O_3$@ZnS-$S_d$, aside from the peaks ascribed to $S^{2-}$ (161.7 eV and 162.9 eV)[40] and $SO_4^{2-}$ (168.4 eV and 169.6 eV)[41], new peaks related to $S_n^{2-}$ (163.4 eV and 164.6 eV)[42] occurred (Fig. 5c). Additionally, the proportion of $S_n^{2-}$ in $Al_2O_3$@CuS-$S_d$ increased from 21.6% to 29.7% (Supplementary Fig. 19, Supplementary Table 5), demonstrating the consistency of the sulfur chemical state on chalcophile metal sulfides. However, in comparison, $Al_2O_3$-$S_d$ featured its characteristic peaks of S 2$p_{3/2}$ at binding energies of 164.0 eV and 167.8 eV (Supplementary Fig. 20), which were ascribed to $S_8^0$ and adsorbed unreacted $SO_2$ species, respectively[43]. Moreover, the binding energy of $Zn^{2+}$ 2$p_{3/2}$ in $Al_2O_3$@ZnS-$S_d$ shifted from 1022.0 eV to 1021.8 eV after $S_d^0$ deposition (Fig. 5d), indicating the formation of unsaturated coordination environments[44]. To better observe the microscopic changes and the dynamic evolution of deposited $S_d^0$, pure ZnS was further synthesized in the same way without adding $Al_2O_3$ pellets to directly serve as support for S-CVD process. The transmission electron microscope (TEM) images of ZnS-$S_d$ showed a decrease in the contrasts of zinc atoms (Supplementary Fig. 21), indicating the formation of Zn defects[19]. The XAFS S L-edge spectra confirmed the formation of $S_n^{2-}$ species in ZnS-$S_d$ (Supplementary Fig. 22a). The extended XAFS (EXAFS) Zn K-edge spectra further revealed a decrease from 3.9 to 3.5 in the coordination number of Zn to S atoms in ZnS-$S_d$ compared to that in pristine ZnS (Supplementary Fig. 22b and c, and Supplementary Table 6), further certifying the formation of unsaturated coordination sites. The XPS depth profiling results depicted that with the Ar$^+$ etching depth increased to 6 mm, the average valence state of $S_n^{2-}$ decreased, suggesting a shortening of the $S_n^{2-}$ chain length close to the ZnS surface (Supplementary Fig. 23). Furthermore, in-situ Raman spectra revealed conversion of $S_8$ to $S_n^{2-}$ in ZnS-$S_d$ at elevated temperatures (Supplementary Fig. 24), which is also demonstrated by Fourier transform infrared spectroscopy (FTIR) and $^{13}C$ nuclear magnetic resonance (NMR) using propylene as an indicator (Supplementary Fig. 25). Therefore, these imply that $S_d^0$ does not simply physically accumulate on the $Al_2O_3$@ZnS surface in form of $S_8^0$ (Eq. 3), but can be activated by Zn atoms and generated chemically bonded $S_n^{2-}$ ($S_d^*$) with unsaturated coordination environments (Eq. 4):

$$Al_2O_3 + S_d^0 \rightarrow Al_2O_3\text{-}S_d^0 \rightarrow Al_2O_3\text{-}S_8^0 \qquad (3)$$

$$Al_2O_3@ZnS + S_d^0 \rightarrow Al_2O_3@ZnS\text{-}S_d^* \qquad (4)$$

The kinetic simulation revealed that the $Hg^0$ adsorption over $Al_2O_3$@ZnS-$S_d$ closely followed the pseudo-first-order kinetic model (Supplementary Fig. 26), emphasizing the important role of the external surface area for $Hg^0$ adsorption progress. Notably, after $Hg^0$ adsorption, there brought out the crystal of β-HgS (JCPDS no. 75-1538), which characterized its main diffraction peaks at 26.4°, 30.6°, 43.8°, and 51.9°, in the XRD pattern of spent $Al_2O_3$@ZnS-$S_d$ (Fig. 5a). The Raman spectrum of spent $Al_2O_3$@ZnS-$S_d$ also verified the formation of Hg−S bonds that located at 249.6 cm$^{-1}$ and 343.6 cm$^{-1}$ (Fig. 5b)[45]. Besides, the proportion of $S_n^{2-}$ species in XPS spectra of spent $Al_2O_3$@ZnS-$S_d$ decreased from 36.7% to 3.0%, while the proportion of

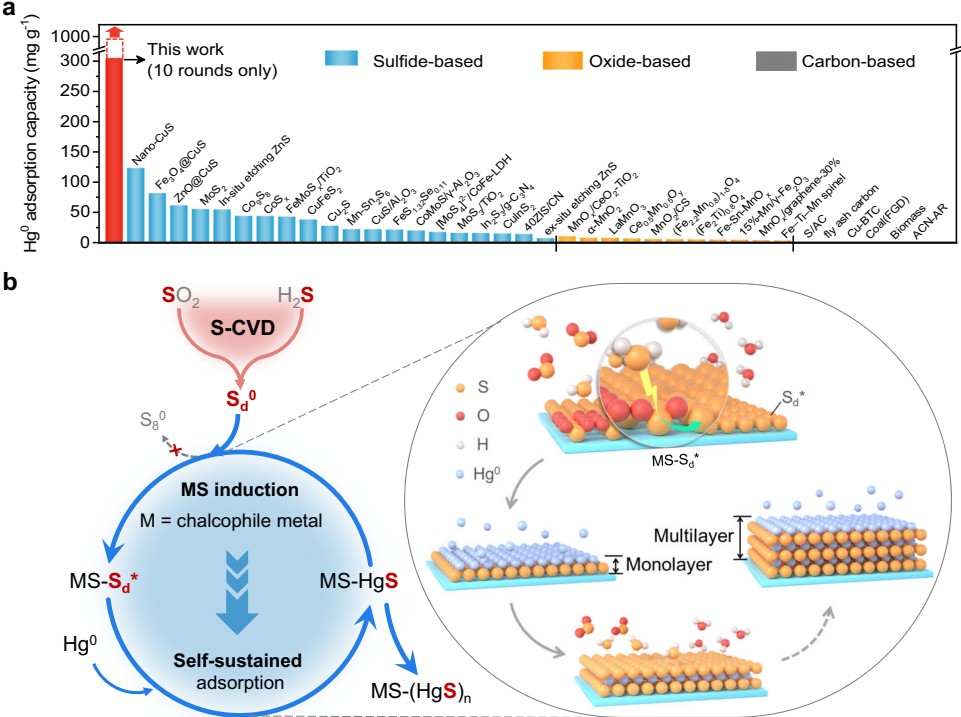

**Fig. 6 | Performance comparison and mechanism schematic. a** Comparison of $Hg^0$ adsorption capacities between $Al_2O_3@ZnS-S_d$ and other reported adsorbents. The raw data can be found in Supplementary Table 7. **b** Schematic illustration of the proposed in-situ S-CVD technology for self-sustained and multilayer adsorption of $Hg^0$. $S_d^0$ is deposited and activated to $S_d^*$ on the $Al_2O_3@MS$ by the reaction between flue gas $SO_2$ and intermittently added $H_2S$; when $S_d^*$ is consumed by $Hg^0$, it can be replenished by repeated S-CVD, resulting in self-sustained and multilayer adsorption of $Hg^0$.

$S^{2-}$ increased from 35.4% to 82.9% (Fig. 5c, Supplementary Table 5). Moreover, the Hg 4f spectrum of spent $Al_2O_3@ZnS-S_d$ exhibited the characteristic peaks of $Hg^{2+}$ $4f_{7/2}$ and $4f_{5/2}$ centering at 100.3 eV and 104.3 eV, respectively (Supplementary Fig. 27). This signifies that adsorbed $Hg^0$ can combine with the active $S_d^*$ on $Al_2O_3@ZnS-S_d$ to form HgS:

$$Al_2O_3@ZnS\text{-}S_d^* + Hg^0 \rightarrow Al_2O_3@ZnS\text{-}HgS \qquad (5)$$

As the 24 h breakthrough rate increased and stabilized with increasing S-CVD rounds, and HgS crystal was observed in the spent $Al_2O_3@ZnS-S_d$, we suspect that the adsorbent surface was gradually covered by produced HgS, which may also possess the ability to activate $S_d^0$ owing to the chalcophile nature of Hg element[46]. Upon this, $Al_2O_3@HgS$ was synthesized and used to testify the performance of $Al_2O_3@HgS-S_d$ for $Hg^0$ adsorption. As shown in Fig. 5e, $Al_2O_3@HgS-S_d$ exhibited an initial $Hg^0$ removal efficiency of ~96% and an 24 h breakthrough ratio of ~80% in each round of experiment, which is close to that of $Al_2O_3@ZnS-S_d$ at the tenth round. As shown in Fig. 5f, compared with $Al_2O_3@HgS$, the XPS S 2p spectrum of $Al_2O_3@HgS-S_d$ brought out new sulfur species located at 163.7 eV and 164.9 eV, which were between the binding energies of $S_n^{2-}$ in $Al_2O_3@ZnS-S_d$ and $S_8^0$ in $Al_2O_3$-$S_d$. This indicates that the average chemical valence of $S_d^*$ on $Al_2O_3@HgS-S_d$ was slightly higher than that on $Al_2O_3@ZnS-S_d$, explaining the decrease in $Hg^0$ adsorption on $Al_2O_3@ZnS$ with increasing S-CVD rounds. The Hg 4f spectra showed that the locations of $Hg^{2+}$ in $Al_2O_3@HgS-S_d$ shifted to lower binding energies compared to those of $Al_2O_3@HgS$ (Supplementary Fig. 28). Moreover, the relationship between the $Q_i$ of $Al_2O_3@HgS-S_d$ and the Hg−S bond energy likewise fell within the negative correlation trend of chalcophile MS (Fig. 2c, Supplementary Fig. 29, Supplementary Table 2).

Notably, as the segmented linear fitting results in Fig. 5g present, with the reaction proceeded, the adsorption rate of $Al_2O_3@ZnS-S_d$ at 120 °C decreased from 1.6 mg $g^{-1}$ $h^{-1}$ to 1.1 mg $g^{-1}$ $h^{-1}$, which converged to that of $Al_2O_3@HgS-S_d$ (1.1 mg $g^{-1}$ $h^{-1}$). The adsorption rate of $Al_2O_3@CuS-S_d$ at 60 °C decreased from 1.5 mg $g^{-1}$ $h^{-1}$ to 1.4 mg $g^{-1}$ $h^{-1}$, which was also converged to that of $Al_2O_3@HgS-S_d$ (1.3 mg $g^{-1}$ $h^{-1}$) (Supplementary Fig. 30). This validates the gradual transition of $S_d^0$ activation and $Hg^0$ adsorption from the raw MS surface to the self-sustained HgS surface, in accordance with the concept presented in Fig. 1b. This verifies the promoting effect of MS on $S_d^0$ is gradually replaced by HgS. Importantly, disregarding the effect of increasing adsorbent size caused by the formation of HgS layer, theoretically, the $Hg^0$ adsorption capacity of $Al_2O_3@ZnS-S_d$ can be continuously increased due to the self-sustained adsorption performance of HgS itself (Fig. 5h). Thus, according to the adsorption rate, the adsorption capacity of $Al_2O_3@ZnS-S_d$ can achieve $(54.2 + 25.1d)$ mg $g^{-1}$ (d > 6 days), thereby breaking the saturation limitations and realizing multilayer adsorption.

DFT calculations were applied to elaborate the $Hg^0$ self-sustained adsorption mechanism on ZnS. ZnS (111) model was established and optimized to investigate the Gibbs free energy ($\Delta G$) of $S_d^0$ activation and the adsorption energy ($E_{ads}$) of $Hg^0$. Given that $S_n^{2-}$ dominated in the $Hg^0$ adsorption process, the $\Delta G$ from $S_8$ ring to $S_n$ ($n$ = 6, 4, 2, and 1) chains were first calculated. As presents in Supplementary Fig. 31, the $ZnS-S_8$ can spontaneously convert to $ZnS-S_6$ and then to $ZnS-S_4$ with a negative $\Delta G$ of −77.9 kJ $mol^{-1}$, while the conversion of $ZnS-S_4$ to $ZnS-S_2$ has a positive $\Delta G$ of 158.6 kJ $mol^{-1}$. This indicates the most stable structure of $ZnS-S_4$. Moreover, compared with ZnS, the Zn−S bond length in $ZnS-S_4$ surface increased from 2.31 Å to 2.38−2.48 Å (Supplementary Fig. 32), in line with the decrease in Zn−S coordination number in $ZnS-S_d$. The $E_{ads}$ of $Hg^0$ adsorption on ZnS, $ZnS-S_8$, and $ZnS-S_4$ were calculated as −46.2, −67.6, and −125.0 kJ $mol^{-1}$, respectively

(Fig. 5i). The highest $E_{ads}$ of ZnS-$S_4$ verifies the important role of $S_4$ chain on $Hg^0$ adsorption. Further, considering the formation of HgS on $Al_2O_3$@ZnS-$S_d$ after self-sustaied adsorption of $Hg^0$, we constructed the ZnS@HgS structure for subsequent $S_d^0$ activation and $Hg^0$ adsorption. The negative $\Delta G$ (−68.4 kJ mol$^{-1}$) from ZnS@HgS-$S_8$ to ZnS@HgS-$S_4$ demostrated its spontaneous conversion process. The $E_{ads}$ of $Hg^0$ on ZnS@HgS-$S_4$ (−73.1 kJ mol$^{-1}$) was higher than those of ZnS@HgS (−26.0 kJ mol$^{-1}$) and ZnS@HgS-$S_8$ (−27.9 kJ mol$^{-1}$). These confirm the role of HgS in the activation of $S_d^0$ and further adsorption of $Hg^0$, supporting the proposed $Hg^0$ self-sustained adsorption mechanism.

## Discussion

Figure 6a and Supplementary Table 7 compare the $Hg^0$ adsorption capacities of $Al_2O_3$@ZnS-$S_d$ with various reported adsorbents. The adsorption capacity of carbon- and oxide-based adsorbents were generally lower than 2 mg g$^{-1}$ and 10 mg g$^{-1}$, respectively, and their performance was severely inhibited by $SO_2$. Among the reported sulfide-based adsorbents, nano-CuS exhibited the highest capacity of 122.4 mg g$^{-1}$; however, it decreased to 89.4 mg g$^{-1}$ in the presence of $SO_2$ and $H_2O$ (ref. 16). Additionally, our previous work found that under scaled-up conditions, ~1 mm $Al_2O_3$@CuS only had the normalized saturated adsorption capacity of 21.0 mg g$^{-1}$ (ref. 41). Another recently reported in-situ acid etching method can boost the adsorption capacity of ZnS to 53.8 mg g$^{-1}$ (ref. 19), but it decreased to 1.4 mg g$^{-1}$ under the scale-up conditions (Supplementary Fig. 33). Impressively, the self-sustained $Al_2O_3$@ZnS-$S_d$ not only reversed the poisoning effect of $SO_2$ but also reached a $Hg^0$ adsorption capacity of 303.9 mg g$^{-1}$ (normalized to ZnS coating amount, and with 24-h breakthrough ratio of ~80%) after 10 rounds of reaction, which is over 250, 60, and 8 times higher than the average of reported carbon-, oxide-, and sulfur-based adsorbents, respectively. Moreover, as the self-sustained adsorption process continues, the $Hg^0$ adsorption capacity of $Al_2O_3$@ZnS-$S_d$ will be increasing, thereby breaking the capacity limitations.

In summary, the $Al_2O_3$@MS-$S_d$ adsorbents assisted with intermittent S-CVD can reverse $SO_2$ poisoning effects and have great potential for efficient and cost-effective $Hg^0$ removal from $SO_2$-containing flue gases. This study demonstrates the crucial role of HgS, whether synthesized or formed by $Hg^0$ adsorption on $Al_2O_3$@MS-$S_d$, in activating of $S_d^0$ into $S_d^*$ (Eqs. 6 and 7). Therefore, this enables the self-sustained adsorption of $Hg^0$ on $Al_2O_3$@MS-$S_d$ surface-like chain reactions (Eqs. 8–10), realizing the multilayer adsorption (Fig. 6b).

$$Al_2O_3@HgS + S_d^0 \rightarrow Al_2O_3@HgS\text{-}S_d^* \tag{6}$$

$$Al_2O_3@MS\text{-}HgS + S_d^0 \rightarrow Al_2O_3(@MS\text{-}HgS)\text{-}S_d^* \tag{7}$$

$$Al_2O_3(@MS\text{-}HgS)\text{-}S_d^* + Hg^0 \rightarrow Al_2O_3@MS\text{-}(HgS)_2 \tag{8}$$

$$Al_2O_3@MS\text{-}(HgS)_2 + S_d^0 \rightarrow Al_2O_3@MS\text{-}(HgS)_2\text{-}S_d^* \tag{9}$$

$$Al_2O_3@MS\text{-}(HgS)_{n-1}\text{-}S_d^* + Hg^0 \rightarrow Al_2O_3@MS\text{-}(HgS)_n \tag{10}$$

## Methods

### Materials
The raw materials used in this work were purchased in chemical purity (>99.5%) from Sinopharm Chemical Reagent Co., Ltd and used without purification. Natural sulfide ores (chalcopyrite and sphalerite) were provided by provided by a non-ferrous smelter in Henan, China.

### Preparation of adsorbents
Different metal sulfides (MS, M = Cu, Zn, Cd, In, Hg, Pb, Sn, Mn, Fe, Co, and Ni) were synthesized via a one-step hydrothermal method as the interface for S-CVD and $Hg^0$ adsorption. According to our previous study, ~1 mm commercial γ-$Al_2O_3$ pellets were chosen as the support for MS (named $Al_2O_3$@MS) to improve gas mass transfer[41]. Taking $Al_2O_3$@ZnS as an example, firstly, 3.50 mL of $H_2O$ containing 2.50 mmol of $ZnSO_4$·$7H_2O$ was added into 4.75 g of γ-$Al_2O_3$ pellets followed by 30 min ultrasonic treatment. Then, the mixture was dried at 60 °C for 9 h to obtain $Al_2O_3$@$ZnSO_4$. After that, the pellets were poured into 50.0 mL of $H_2O$ containing 2.50 mmol of $Na_2S$·$9H_2O$ and transferred to a 100 mL Teflon-lined autoclave. After reacting at 120 °C for 12 h, the resultant $Al_2O_3$@ZnS was collected by vacuum filtration, washed with deionized water and ethanol, and dried at 60 °C in an oven for 12 h. Other $Al_2O_3$@MS adsorbents were synthesized in the same way, except that different metal precursors, including $CuSO_4$·$5H_2O$ (2.50 mmol), $CdCl_2$ (2.50 mmol), $InCl_3$·$4H_2O$ (1.67 mmol), $HgCl_2$ (2.50 mmol), $(CH_3COO)_2Pb$ (2.50 mmol), $SnCl_2$ (2.50 mmol), $MnSO_4$·$H_2O$ (2.50 mmol), $FeCl_3$ (1.67 mmol), $CoCl_2$·$6H_2O$ (2.50 mmol), and $NiCl_2$·$6H_2O$ (2.50 mmol), were used instead of $ZnSO_4$·$7H_2O$.

### In-situ low-temperature sulfur chemical vapor deposition
The in-situ S-CVD was conducted in a self-made fixed-bed reactor system as shown in Supplementary Fig. 1. To prevent the pre-reaction between $H_2S$ and $SO_2$ in the pipeline (heat protection at 120 °C), a three-way quartz reaction tube was used to introduce different gas components, of which the main tube (10.0 mm inner diameter) was used to feed 5000 ppm $SO_2$, and the side tube (4.0 mm inner diameter) embedded in the main reaction tube was used to feed 100 ppm $H_2S$. The $H_2S$ concentration was detected by a gas chromatograph (Agilent GC 8860) equipped with a flame photometric detector. To verify the feasibility of the S-CVD strategy, we used $SiO_2$ wafer as substrate to conduct the reaction. The SEM images of $SiO_2$-$S_d$ exhibited a homogeneous surface with a few grooves and the ESD mapping images showed the uniform distribution of S element (Supplementary Fig. 34a–f). Moreover, the SEM image of the cross-section of $SiO_2$-$S_d$ showed the formation of a deposition layer (Supplementary Fig. 34g, h).

Specifically, in this work, single S-CVD (15 min) was used to evaluate the enhanced effect of S-CVD on the activities of different $Al_2O_3$@MS adsorbents, while $Al_2O_3$@MS or natural sulfide ores treated with intermittent S-CVD (30 min each round) were applied to assess their self-sustained adsorption performance for $Hg^0$.

### Characterizations
The chemical composition of the as-prepared adsorbents was analyzed by X-ray fluorescence spectroscopy on an Epsilon 3X instrument (Netherlands). The thermal stability of the adsorbents was analyzed by a thermogravimetric analyzer (NETZSCH, STA 2500 Regulus) at a heating rate of 5 °C min$^{-1}$ from 20 °C to 500 °C in $N_2$ atmosphere. The BET-specific surface area and pore structure were detected by an automatic porosity analyzer apparatus (Quantachrome, Autosorb-iQ, USA). The XRD patterns were obtained on Shimadzu XRD-6100 (Japan) with Cu Kα radiation (scan speed: 8° min$^{-1}$, and 2θ range of 10−80°). The SEM images of adsorbents were performed on ZEISS Sigma 300 (Germany), and EDS/mapping images using Oxford Xplore 30 (UK). XPS spectra were determined on an X-ray photoelectron spectrometer (Thermo Scientific ESCALAB Xi+, USA) equipped with a mono Al Kα X-ray source. TEM images were carried out on FEI Tecnai F20. The XPS depth profiling is measured by Ar$^+$ ion etching material surface layers at different depths. XAFS spectra of S L-edge and Zn K-edge were detected by synchrotron radiation light sources (see detail in Supplementary Methods 1). Raman spectra were measured on a Raman spectrometer (Horiba LabRAM HR Evolution, Japan) with 532 nm line of Ar$^+$ laser for excitation. FTIR spectra were recorded on a Fourier-

Transform infrared spectrometer (Nicolet 6700, USA) using the potassium bromide pellet technique. $^{13}$C NMR spectra were performed on a nuclear magnetic resonance spectrometer (Bruker Avance Neo 400WB, Germany).

## Gaseous mercury adsorption assessment

$Hg^0$ adsorption performances of as-prepared adsorbents were assessed in the same reactor system. The $Hg^0$ vapor was obtained by an $Hg^0$ permeation device (VICI Metronics) loaded in a U-shaped glass tube. The $H_2O$ vapor was produced by a steam generator. The concentrations of both $Hg^0$ and $H_2O$ were controlled by adjusting the water bath temperature and carrier gas ($N_2$) flow rate. Other gas component including $O_2$, $SO_2$, and NO, were obtained directly from compressed gas in cylinders. The gas components of the simulated flue gas, including $Hg^0$ (1.5–2.5 mg m$^{-3}$), $SO_2$ (5000–60,000 ppm, when used), and $H_2O$ (4%, when used), $O_2$ (5%, when used), and NO (100 ppm, when used) were mixed evenly before entering the fixed-bed reactor. The total flow rate was controlled at 300–360 mL min$^{-1}$, the adsorbent mass used in each experiment was 0.3–1.0 g. The detailed experiment conditions were summarized in Supplementary Table 8. Lumex RA 915+ was used to record the inlet and outlet $Hg^0$ concentrations and before each experiment, the inlet concentration should remain stable (±0.05 μg m$^{-3}$) for more than 10 min. The exhaust gas was absorbed by 0.1 mol L$^{-1}$ potassium permanganate solution and activated carbon before discharged.

The $Hg^0$ removal efficiency ($\eta$, %) and $Hg^0$ adsorption capacity ($Q$, mg·g$^{-1}$) were calculated according to following equations:

$$\eta = \frac{Hg^0_{in} - Hg^0_{out}}{Hg^0_{in}} \times 100\% \tag{11}$$

$$Q = \int_{t_1}^{t_2} \frac{Hg^0_{in} - Hg^0_{out}}{m} \times f \, dt \tag{12}$$

where $Hg^0_{in}$ and $Hg^0_{out}$ (mg m$^{-3}$) are the inlet and outlet $Hg^0$ concentrations, respectively, $m$ (g) is the adsorbent mass, $f$ (mL min$^{-1}$) represents the total flow rate, and $t$ (min) donates the adsorption time.

$Hg^0$-TPD experiments were conducted to identify the $Hg^0$ species adsorbed on the adsorbent surface and its stability. A certain amount of spent adsorbent was heated from 50 to 450 °C with a heating rate of 5 °C min$^{-1}$ in pure $N_2$ to desorb the adsorbed mercury. The signal of desorbed $Hg^0$ was also detected by Lumex RA 915+.

## Kinetics and theory calculation

$Hg^0$ adsorption kinetic models and theoretical calculations of $S_d^0$ activation and $Hg^0$ adsorption behavior are detailed in Supplementary Methods 2 and 3.

# Data availability

All data supporting the findings of this study are available within the paper and the Supplementary information files or available from the corresponding author upon request.

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

## Acknowledgements

This work was partly supported by the National Key Research and Development Program of China (2022YFC3901100, N.Y.; 2022YFC3703800, H.X.) and the National Natural Science Foundation of China (No. 21976118, N.Y.; No. 52070129, Z.Q.; No. 22176121, H.X.).

## Author contributions

Q.H., H.X., Z.Q., and N.Y. conceived the idea and designed the research; Q.H. and X.S. prepared materials and performed characterizations; J.L. and W.H. assembled the test system; Q.H. and H.X. analyzed and interpreted the results; and Q.H., H.X., Z.Q., L.Z., and N.Y. contributed to the writing and revising of the paper.

## Competing interests

The authors declare no competing interests.
