## [Peer Review File · Nature Communications]

In-situ Low-temperature Sulfur CVD on Metal Sulfides with SO_2 to Realize Self-sustained Adsorption of MercuryREVIEWER COMMENTS

Reviewer #1 (Remarks to the Author):

Mercury pollution is a hot issue in the world. In this work, an in-situ low-temperature sulfur chemical vapor deposition (SCVD) technique that converting flue gas SO₂ and H₂S to deposited sulfur (S_d) is used to realize self-sustained adsorption of mercury. However, the quality and innovation of this manuscript is insufficient, so it is not recommended for publication in Nature Communication.

1. CVD is generally used for surface coating, does S_d form a film on the surface, the size of the film thickness? How to ensure the uniformity of S_d deposition on Al₂O₃@MS surface, in what final form is it combined on the surface, is it adsorption or are new chemical bonds formed? At present, the characterization and interpretation of S structure on the surface of this composite material are relatively weak.

2. As shown in Fig1, is the adsorption of Hg₀ only related to surface S_d and not related to M or S in MS? Is the improvement of the adsorption performance of Hg₀ attributable to the addition of S_d and the combination of MS to form more active S or simply due to the activity of S_d itself? Which elements are activated? Does metal M play a role in this process?

3. After the reaction temperature exceeds 120 °C, will the surface activity S be deactivated and the adsorption performance decrease?

4. The author mentions that S_d is amorphous after deposition on Al₂O₃@MS, and becomes HgS crystal after adsorption of Hg₀. XPS alone is not enough to explain the microscopic reaction process and mechanism, and the author needs to provide more sufficient evidence.

5. After adsorption of Hg₀, the surface of the material is already HgS crystal, the author said, without desorption of Hg, S_d deposition again, can continue to carry out this cycle process, how to prove that layer by layer while the surface of the adsorbent still has enough space for S_d to be added? Is the S_d activated by the S of HgS or the S of MS, or is it the S_d itself?

Reviewer #2 (Remarks to the Author):

The authors developed a novel and efficient in-situ low-temperature SCVD method using SO₂ and H₂S as feed gas to realize self-sustained adsorption of mercury over metal sulfides sorbents, which is verified via large numbers of experimental tests and characterizations. The presented idea and method is very interesting and enlightening, which can contribute to the development of cost-effective techniques or sorbents for mercury emission control. Thus, a minor revision is recommended.

1. Line 71-73, pertinent literature should be added to support the role of metal sites of MS in activating S_d into Sn₂- as this is very important to testify or interpret the major reaction scheme of the proposed SCVD method.

2. Line 77-78 and line 315, the authors claimed that reaction product HgS can also serve as a fresh interface for S_d deposition and activation. Likewise, is there any literature or theory to support this

standpoint?

3. Line 146, the sentence...chalcophile metals have a higher tendency to form highly covalent bonds with sulfur, making them more effective for Hg₀ adsorption^{35,36}." Several scholars had claimed that the weak strength of the M-S bond is favorable for Hg₀ adsorption (The sequestration of gaseous Hg₀ by sphalerite with Fe substitution structure-activity relationship. *J. Phys. Chem. C* 2019, 123, 2828–2836.; Outstanding performance of ZnS/TiO₂ for the urgent disposal of liquid mercury leakage indoors: Novel support effect, reaction mechanism and kinetics. *J. Hazard. Mater.* 403 (2021) 123867), which is contrary to the authors' viewpoint. What is the reason?

4. H₂S is a toxic exhaust gas, thus, the use of H₂S as feed gas is also beneficial in terms of environment protection and waste recycling, this could be mentioned in the introduction section. Moreover, SO₂ is rich in coal-fired flue gas, however, H₂S is very few. Thus, the sources of H₂S should be indicated or discussed.

5. Line 171-172, besides the decomposition of partially formed mercury compounds, is there any other reason for the decrease of Hg₀ removal performance at 140 °C? Based on Hg-TPD results (Supplementary Fig. 6), it is clear that mercury release only kicks off at 150 °C. Thus, very few formed mercury compounds may decompose at 140 °C. In my opinion, the Claus reaction is significantly affected by reaction temperature. With the increase of temperature, the chain or ring structure of S_x products may also change, possibly leading to decreased Hg₀ removal performance.

6. Line 196-197, the reasons for the positive effect of SO₂ on Hg₀ removal could be discussed more.

7. Line 237, the peak at 164.6 is probably owing to elemental sulfur S₈, as confirmed by Raman spectra in Fig. 3b, S₈ is also present on Al₂O₃@ZnS-Sd.

8. How to prepare Al₂O₃@HgS sorbent? Please add this information in Methods section as well.

9. Line 176-177, the reasons for the negative effect of H₂O on mercury adsorption should be discussed more. As reported, steam shows a complicated influence on Hg₀ capture over metal sulfides or sulfur-bearing materials (Novel metal sulfide sorbents for elemental mercury capture in flue gas: A review. *Fuel* DOI: 10.1016/j.fuel.2023.129829), insignificant, promotional, suppressive effects had all been reported.

10. Supplementary Table 4, the breakthrough percent for each sorbent could be provided for facile comparison. Line 24 and 359, the breakthrough percent should be specified as well.

11. Fig 1b and Supplementary Fig. 3, 7,13, and 15, it is suggested to add the reaction conditions (gas composition, reaction temperature, sorbent dosage, adsorption time) in the figure captions to make the figures more readable. Specially, the breakthrough percents could be offered in Fig. 1b. Because Fig. 1b is the crucial data to testify the idea of this work and would be valuable for peers to refer to.

12. In Scheme 1, some texts are needed to briefly describe the experimental procedure of SCVD method in the figure caption.

13. In our experimental experience, H₂S could significant disable the Hg₀ detector, such as VM3000. Is it also for Lumex? How did the authors deal with this issue?

14. Line 66, the sentence '...and the species tuning of sulfur...' is confusing, please rephrase it to make the idea clearer.

15. Some typos:

--Line 68, a word 'over or on' may be missing prior to '...metal sulfides'.

--Fig. 3, the figure caption should be corrected, which seems incorrect.

--Line 323-324, '...adsorption performance of with Al₂O₃@HgS-Sd at 60 °C.'

Reviewer #3 (Remarks to the Author):

The removal of flue gas mercury using adsorbents is vital, but current adsorbents face challenges due to complex synthesis, poor resistance to SO₂, and low adsorption capacities. Hong et al. observed an intriguing phenomenon that flue gas SO₂ can directly transform into active sulfur sites on metal sulfide surfaces. Based on this discovery, they proposed an innovative in-situ sulfur CVD method for flue gas mercury capture, ensuring sustained self-sustained mercury adsorption performance. This approach continuously converts harmful flue gas SO₂ into nascent sulfur species and effectively utilizes chalcophile metals to modulate sulfur speciation, achieving prolonged high-capacity mercury adsorption. Therefore, I think it is suitable to be published in Nature Communications after a minor revision. The related comments are as follows:

- 1) In the Abstract section, the full name of Sn²⁺ should be given, and there is more than one expression for Sn²⁺ throughout the manuscript (line 54, line 72), please standardize the expression to avoid confusion.
- 2) Line 68, please give a clear definition of “self-sustained”. What is the relationship between “self-sustained” and “multilayer” adsorption?
- 3) The XPS spectra of Zn 2p should be added in the Supplementary information.
- 4) Considering the application temperature window of the adsorbent, additional experiments are recommended to evaluate the thermal stability and the efficiencies of the adsorbent.
- 5) Figure 4a shows great sustainable Hg⁰ adsorption performance of Al₂O₃@ZnS-Sd after each round of SCVD. I think it is necessary to further investigate the Hg⁰ re-emission after multiple rounds of experiments.
- 6) In Lines 324-326, besides the adsorption rates of Al₂O₃@CuS-Sd, the corresponding self-sustained adsorption curve should be provided in the Supplementary information as well.
- 7) The synthetic method of in-situ etching ZnS and Al₂O₃@ZnS in Supplementary Fig. 15 should be added.
- 8) The paragraph in Lines 364-373 only discusses the applicability of Al₂O₃@MS-Sd in smelting flue gas, while in the Abstract, it is mentioned that it “enables efficient on-site mercury removal in various scenarios”. Please add the discussion of its applicability in other scenarios or revise the statement in the Abstract.
- 9) Lines 441-451, the references to different kinetic models should be cited.

The response to the reviewers' comments are as follows:

REVIEWER COMMENTS

Reviewer #1 (Remarks to the Author):

Mercury pollution is a hot issue in the world. In this work, an in-situ low-temperature sulfur chemical vapor deposition (SCVD) technique that converting flue gas SO_2 and H_2S to deposited sulfur (S_d) is used to realize self-sustained adsorption of mercury. However, the quality and innovation of this manuscript is insufficient, so it is not recommended for publication in Nature Communications.

1. CVD is generally used for surface coating, does S_d form a film on the surface, the size of the film thickness? How to ensure the uniformity of S_d deposition on $\text{Al}_2\text{O}_3@\text{MS}$ surface, in what final form is it combined on the surface, is it adsorption or are new chemical bonds formed? At present, the characterization and interpretation of S structure on the surface of this composite material are relatively weak.

Response: Thanks for your comments. To verify the formation of S_d film during the SCVD process, we further supplemented the SCVD reaction on SiO_2 wafer. The SEM images of $\text{SiO}_2\text{-S}_d$ exhibited a uniform surface without visible agglomeration (Supplementary 26a-c) and the corresponding ESD mapping images showed the uniform distribution of S element (inset in Supplementary 26c, Supplementary 26d-f). Moreover, the SEM image of the cross section of $\text{SiO}_2\text{-S}_d$ showed a 6 μm thick deposition layer (Supplementary 26g and h). This certifies the feasibility of the SCVD strategy.

Supplementary Fig. 26 (a-c) SEM images of the SiO₂-S_d surface. (d-f) EDS mapping images corresponding to (b). SEM images of the cross section of (g) SiO₂ and (h) SiO₂-S_d.

Besides, the repeatability results of Raman spectra measured at different points exhibited characteristic vibration modes with the same Raman shifts and intensities. This further confirms the uniformity of S_d deposition on Al₂O₃@MS surface.

Figure R1. Repeatability results of Raman spectra of Al₂O₃@ZnS-S_d measured at different points.

Although the SCVD strategy produced elemental sulfur (S_d) on Al₂O₃@MS surface, the interaction between S_d and interfacial MS can change the final speciation of S_d, which has been demonstrated through a series of characterizations. Taking Al₂O₃@ZnS as example, in this work, we found that the Raman spectrum of Al₂O₃@ZnS-S_d exhibited the vibration modes of S-S bonds, but its Raman shifts were higher than those of elemental sulfur S₈ (Fig. 3b). Further, the XPS results showed that S_n²⁻ is formed in Al₂O₃@ZnS-S_d rather than S⁰ that formed on Al₂O₃-S_d (Fig. 3c and Supplementary Fig.

14). We have supplemented the Zn 2*p* XPS spectra of Al₂O₃@ZnS and Al₂O₃@ZnS-S_d, which showed that the location of Zn²⁺ species shifted to lower binding energies after S_d deposition on Al₂O₃@ZnS (Fig. 3d). These indicates the deposited S_d can be activated by the metal sites (Zn atoms) and combined with S sites to form chemically bonded S_n²⁻ with unsaturated coordination environments.

Fig. 3. **a** XRD patterns of Al₂O₃@ZnS, Al₂O₃@ZnS-S_d, and spent Al₂O₃@ZnS-S_d. **b** Raman spectra of Al₂O₃@ZnS, Al₂O₃@ZnS-S_d, spent Al₂O₃@ZnS-S_d, and pure S₈. **c** S 2*p* and **d** Zn 2*p* XPS spectra of Al₂O₃@ZnS, Al₂O₃@ZnS-S_d, and spent Al₂O₃@ZnS-S_d.

The supplemented contents have been revised in the manuscript as follows:

Revised:

Line 464-468: “To verify the feasibility of the SCVD strategy, we first used SiO₂ wafer as substrate to conduct the reaction. The SEM images of SiO₂-S_d exhibited a homogeneous surface with a few grooves and the ESD mapping images showed the uniform distribution of S element (Supplementary 26a-f). Moreover, the SEM image of the cross section of SiO₂-S_d showed the formation of a deposition layer (Supplementary

26g and h).”

Line 291-299: “Moreover, the deposition of S_d also influenced the speciation of Zn element in Al₂O₃@ZnS. The binding energy of Zn²⁺ 2p_{3/2} in Al₂O₃@ZnS shifted from 1022.0 to 1021.8 eV after S_d deposition (Fig. 3d), indicating the formation of unsaturated coordination environments⁵¹. Thus, this implies that S_d does not simply physically accumulate on the Al₂O₃@ZnS surface as elemental state, but can be activated by the metal sites (Zn atoms) and combined with S sites to form chemically bonded S_n²⁻ with unsaturated coordination environments (Eq. 3):

where [] represents the unsaturated coordination.”

Reference:

51 Hao, X. *et al.* Zinc vacancy-promoted photocatalytic activity and photostability of ZnS for efficient visible-light-driven hydrogen evolution. *Appl. Catal. B-Environ.* **221**, 302-311, doi:<https://doi.org/10.1016/j.apcatb.2017.09.006> (2018).

2. As shown in Fig1, is the adsorption of Hg⁰ only related to surface S_d and not related to M or S in MS? Is the improvement of the adsorption performance of Hg⁰ attributable to the addition of S_d and the combination of MS to form more active S or simply due to the activity of S_d itself? Which elements are activated? Does metal M play a role in this process?

Response: Thanks for your comment. As shown in Fig. 1b, different fresh Al₂O₃@MS adsorbents (blue bars) all showed a certain adsorption activity for Hg⁰, which is in consistent with our previous researches (Recyclable CuS sorbent with large mercury adsorption capacity in the presence of SO₂ from non-ferrous metal smelting flue gas. *Fuel* 2019, 235, 847-854; Heterogeneous Reaction Mechanisms and Functional Materials for Elemental Mercury Removal from Industrial Flue Gas. *ACS ES&T Engineering*, 2021, 1, 1383-1400). However, the active sites in raw MS are limited, making it susceptible to adsorption saturation, especially in the case of scaled-up experiments (Adsorption of Gaseous Mercury for Engineering Optimization: From

Macrodynamics to Adsorption Kinetics and Thermodynamics. *ACS ES&T Engineering*, 2021, 1, 865-873). Thus, in this work, we further proposed this novel SCVD strategy to supplement new S_d sites on MS for Hg^0 adsorption after the original sites were depleted.

Fig. 1. b Hg^0 adsorption capacities of different $Al_2O_3@MS$ and $Al_2O_3@MS-S_d$ ($M = Cu, Zn, Cd, In, Pb, Sn, Ni, Co, Fe,$ and Mn). Reaction conditions: sorbent mass = 0.3 g, total flow rate = 360 mL min^{-1} , SO_2 concentration = 5000 ppm (during SCVD process), H_2S concentration = 100 ppm (during SCVD process), SCVD time = 15 min, Hg^0 concentration = $(1.5 \pm 0.05)\text{ mg m}^{-3}$, reaction temperature = $80\text{ }^\circ\text{C}$, and reaction time = 180 min.

The improvement of the adsorption performance of Hg^0 is attributable to the addition of S_d and the combination of MS to form more active S rather than the activity of S_d itself. As shown in Supplementary Fig. 5, the $Al_2O_3-S_d$, $Al_2O_3@ZnSO_4-S_d$, and $Al_2O_3@Na_2S-S_d$ did not show improved performance compared to their raw materials. This implies that S_d itself is inactive and the isolated presence of metal or sulfur sites cannot directly activate the deposited S_d . However, as shown in the red bars in Fig. 1b, different $Al_2O_3@MS-S_d$ exhibited substantial differences in enhancing their Hg^0 adsorption capacities compared to corresponding $Al_2O_3@MS$. Among them, only chalcophile metal sulfides, like CuS, ZnS, CdS, In_2S_3 , PbS, and SnS, showed great enhancement after SCVD process. This indicates the significant role of metal site in MS on the activation of S_d .

Supplementary Fig. 5 Hg^0 adsorption performance of raw and activated Al_2O_3 , $\text{Al}_2\text{O}_3@ZnSO_4$, and $\text{Al}_2\text{O}_3@Na_2S$. Reaction conditions: sorbent mass = 0.3 g, SO_2 concentration = 5000 ppm (during SCVD process), H_2S concentration = 100 ppm (during SCVD process), SCVD time = 15 min, Hg^0 concentration = $(1.5 \pm 0.05) \text{ mg m}^{-3}$, reaction temperature = $80 \text{ }^\circ\text{C}$, and total flow rate = 360 mL min^{-1} .

Hence, to quantitatively explore the role of metal sites in MS on the activation of S_d , we further construct the tendency between the Hg^0 adsorption capacity increment ($Q_i = Q_{\text{Al}_2\text{O}_3@MS-S_d} - Q_{\text{Al}_2\text{O}_3@MS}$) and metal-sulfur (M-S) bond energy. The results showed that the Q_i of chalcophile metal sulfides showed a negative relationship with the increase of M-S bond length. However, for siderophile metal sulfides, higher M-S bond lengths instead led to relatively higher Q_i . It can be deduced that the activity of S_d on sulfide interface highly depends on the geochemical characteristics of metal element and the M-S affinity.

Supplementary Fig. 6 The tendency between the adsorption capacity increment and the metal-sulfur bond length of different metal sulfides.

Supplementary Table 3 Structural characterization of the crystalline form of different metal sulfides¹.

Material	Crystal system	Space group symbol	M-S bond length (Å)
CuS	Hexagonal	$P6_3/mmc$	2.18, 2.28, 2.35
ZnS	Cubic	$F\bar{4}3m$	2.33

HgS	Cubic	$F\bar{4}3m$	2.57
CdS	Cubic	$Fm\bar{3}m$	2.75
In_2S_3	Trigonal	$\bar{R}3c$	2.57, 2.71
PbS	Cubic	$Fm\bar{3}m$	2.98-3.00
SnS	Cubic	$Fm\bar{3}m$	2.91
Fe_2S_3	Trigonal	$\bar{R}3c$	2.22, 2.31
CoS	Hexagonal	$P6_3/mmc$	2.32
MnS	Cubic	$Fm\bar{3}m$	2.14
NiS	hexagonal	$P6_3/mmc$	2.37

Furthermore, taking $Al_2O_3@ZnS$ as example, in this work, we found that S_n^{2-} is formed in $Al_2O_3@ZnS-S_d$ rather than S^0 that formed on $Al_2O_3-S_d$ through XPS characterization (Fig. 3c and Supplementary Fig. 14). Moreover, the Zn 2p XPS spectrum in $Al_2O_3@ZnS-S_d$ shifted to lower binding energies compared to $Al_2O_3@ZnS$ (Fig. 3d), indicating the formation of unsaturated coordination environments. Thus, the improvement of the Hg^0 adsorption performance is mainly attributed the activation of S_d into S_n^{2-} by the chalcophile metal sites in MS.

Fig. 3. c S 2p and d Zn 2p XPS spectra of $Al_2O_3@ZnS$, $Al_2O_3@ZnS-S_d$, and spent $Al_2O_3@ZnS-S_d$.

Supplementary Fig. 14 S 2p XPS spectrum of $Al_2O_3-S_d$.

The related contents have been added in the revised manuscript as follows:

Revised:

Line 165-173: “Additionally, to quantitatively explore the role of metal sites in MS on the activation of S_d, we further construct the tendency between the Hg⁰ adsorption capacity increment ($Q_i = Q_{Al_2O_3@MS-S_d} - Q_{Al_2O_3@MS}$) and metal-sulfur (M–S) bond energy (measured by bond length³⁹, Supplementary Table 3) of the investigated metal sulfides. As depicted in Supplementary Fig. 6, the Q_i of chalcophile metal sulfides showed a negative relationship with the increase of M–S bond length. However, for siderophile metal sulfides, higher M–S bond lengths instead led to relatively higher Q_i . It can be deduced that the activity of S_d on sulfide interface highly depends on the geochemical characteristics of metal element and the M–S affinity.”

Line 291-299: “Moreover, the deposition of S_d also influenced the speciation of Zn element in Al₂O₃@ZnS. The binding energy of Zn²⁺ 2p_{3/2} in Al₂O₃@ZnS shifted from 1022.0 to 1021.8 eV after S_d deposition (Fig. 3d), indicating the formation of unsaturated coordination environments⁵¹. Thus, this implies that S_d does not simply physically accumulate on the Al₂O₃@ZnS surface as elemental state, but can be activated by the metal sites (Zn atoms) and combined with S sites to form chemically bonded S_n²⁻ with unsaturated coordination environments (Eq. 3):

where [] represents the unsaturated coordination.”

References:

- 39 Jain, A. *et al.* Commentary: The Materials Project: A materials genome approach to accelerating materials innovation. *APL Mater.* **1**, 011002, doi:10.1063/1.4812323 (2013).
- 51 Hao, X. *et al.* Zinc vacancy-promoted photocatalytic activity and photostability of ZnS for efficient visible-light-driven hydrogen evolution. *Appl. Catal. B-Environ.* **221**, 302-311, doi:https://doi.org/10.1016/j.apcatb.2017.09.006 (2018).

3. After the reaction temperature exceeds 120 °C, will the surface activity S be deactivated and the adsorption performance decrease?

Response: Thanks for your comment. As shown in Fig. 2b, we have supplemented the

performance of $\text{Al}_2\text{O}_3@\text{ZnS-S}_d$ at temperatures of 140 and 160 °C. The results showed that the Hg^0 removal efficiency of $\text{Al}_2\text{O}_3@\text{ZnS-S}_d$ decreased to 86.6% (140 °C) and 75.9% (160 °C) within 180 min. This is presumably due to the re-decomposition of partially captured Hg^0 (Supplementary Fig. 9) or the decreased Hg^0 affinity caused by the change in S_d speciation at high temperatures (Supplementary Fig. 10). Moreover, the SCVD strategy is mainly designed for the Hg^0 removal from flue gas during or after the dynamic wave scrubber, where the temperature range is between 60 and 100 °C, in non-ferrous smelting process (Reverse Conversion Treatment of Gaseous Sulfur Trioxide Using Metastable Sulfides from Sulfur-Rich Flue Gas. *Environ. Sci. Technol.* 2022, 56, 10935-10944). Thus, $\text{Al}_2\text{O}_3@\text{ZnS-S}_d$ is suitable for this temperature range.

The related contents have been revised as follows:

Revised:

In manuscript:

Line 195-201: “Fig. 2b and Supplementary Fig. 8 illustrate that increasing in adsorption temperature from 60 to 120 °C improved the Hg^0 removal efficiency of $\text{Al}_2\text{O}_3@\text{ZnS-S}_d$ from 23.8% to 89.9% (within 180 min). However, when temperature was further elevated to 140 °C and 160 °C, its activity dropped to 86.6% and 75.9%, respectively, presumably due to the re-decomposition of partially captured Hg^0 (Supplementary Fig. 9) or the decreased Hg^0 affinity caused by the change in S_d speciation at high temperatures (Supplementary Fig. 10).”

Fig. 2. b The effects temperature and gas components on Hg^0 removal efficiency of

activated $\text{Al}_2\text{O}_3@\text{ZnS}$.

In Supplementary Information:

Supplementary Fig. 8 Hg^0 adsorption curves of $\text{Al}_2\text{O}_3@\text{ZnS-S}_d$ at different temperatures. Reaction conditions: sorbent mass = 0.3 g, SO_2 concentration = 5000 ppm (during SCVD process), H_2S concentration = 100 ppm (during SCVD process), SCVD time = 15 min, Hg^0 concentration = $(1.5 \pm 0.05) \text{ mg m}^{-3}$, and total flow rate = 360 mL min^{-1} .

Supplementary Fig. 9 Hg^0 Temperature programmed desorption (Hg^0 -TPD) curves of spent $\text{Al}_2\text{O}_3@\text{ZnS-S}_d$ after Hg^0 adsorption with or without the presence of SO_2 .

Supplementary Fig. 10 Raman spectra of ZnS and in-situ Raman spectra of ZnS-S at temperature range of 25-140 °C.

“To explore the dynamic variation of sulfur species in $\text{Al}_2\text{O}_3@\text{ZnS-S}_d$. Pure ZnS was synthesized to directly serve as support for SCVD process. The in-situ heating Raman spectra of the resulting ZnS-S_d showed that with temperature increased from 25 to 80 and further to 140 °C, the intensities of the characteristic peaks of S₈ gradually decreased, while the vibration modes of S_x chain increased¹. Yang et al. found that long-chain S_x had negligible adsorption ability, while short-chain S_x had a high affinity for Hg⁰ (ref. ²). As shown, the ratio of short-chain S₃₋₅ (465 cm⁻¹) to long-chain S₆₋₈ (414 cm⁻¹) decreased from 1.40 at 80 °C to 0.93 at 140 °C³. This could also explain the decrease in Hg⁰ adsorption activity at temperatures above 120 °C.”

SI References:

- 1 Li, Y. *et al.* Photoreduction of inorganic carbon (+IV) by elemental sulfur: Implications for prebiotic synthesis in terrestrial hot springs. *Sci. Adv.* **6**, eabc3687, doi:10.1126/sciadv.abc3687 (2020).
- 2 Yang, Y. *et al.* Different Crystal Forms of ZnS Nanomaterials for the Adsorption of Elemental Mercury. **55**, 6965-6974, *Environ. Sci. Technol.*, doi:10.1021/acs.est.0c05878 (2021).
- 3 Xu, W. *et al.* Fundamental mechanistic insights into the catalytic reactions of Li—S redox by Co single-atom electrocatalysts via operando methods. *Sci. Adv.* **9**, eadi5108, doi: 10.1126/sciadv.adi5108 (2023).

4. The author mentions that S_d is amorphous after deposition on $Al_2O_3@MS$, and becomes Hg^0 crystal after adsorption of Hg^0 . XPS alone is not enough to explain the microscopic reaction process and mechanism, and the author needs to provide more sufficient evidence.

Response: Thanks for your suggestion. To explore the SCVD and Hg^0 adsorption mechanism, we have also provided characterization results of XRF, ESD, XRD, and Hg^0 -TPD in the original manuscript, and these results were in consistent with that of XPS. According to this comment, we further supplemented the Raman spectra to confirm the reaction mechanism.

For the SCVD reaction process: i) The XRF (Supplementary Fig. 4 and Supplementary Table 1) and EDS (Supplementary Fig. 18) results showed that increase sulfur content of $Al_2O_3@ZnS-S_d$ compared to $Al_2O_3@ZnS$; ii) The Raman spectrum of $Al_2O_3@ZnS-S_d$ exhibited the vibration modes of S-S bonds, but their Raman shifts were higher than those of elemental sulfur S_8 (Fig. 3b); iii) No diffraction peaks related to elemental sulfur emerged in the XRD pattern of $Al_2O_3@ZnS-S_d$ (Fig. 3a); iv) The XPS spectra showed the formation of S^0 in $Al_2O_3-S_d$ (Supplementary Fig. 14) and S_n^{2-} in $Al_2O_3@ZnS-S_d$ (Fig. 3c). These evidences suggest the amorphous structure of formed S_d on $Al_2O_3@ZnS$ after SCVD process.

For Hg^0 adsorption mechanism: i) The XRD pattern of spent $Al_2O_3@ZnS-S_d$ presented the diffraction peaks of β - HgS (Fig. 3a); ii) The XPS spectra showed that after Hg^0 adsorption, the proportion of S_n^{2-} in $Al_2O_3@ZnS-S_d$ decreased from 36.7% to 3.0%, while the proportion of S^{2-} increased from 35.4% to 82.9% (Fig. 3c), and the characteristic peaks of Hg^{2+} appeared in the Hg 4f spectrum of spent $Al_2O_3@ZnS-S_d$ (Fig. 3d); iii) The Hg^0 -TPD curve of spent $Al_2O_3@ZnS-S_d$ exhibited the desorption peak centered at 210 °C (Supplementary Fig. 9), which is consistent with the decomposition temperature of β - HgS ; iv) According to the result of ESD, the ratio of Hg to S_d in spent $Al_2O_3@ZnS-S_d$ was calculated as 1.04 (Supplementary Fig. 18), indicating a near 1:1 reaction molar ratio of S_d to Hg^0 . These evidences indicate that the adsorbed Hg^0 can be oxidized by the S_n^{2-} site in $Al_2O_3@ZnS-S_d$ to form stable HgS .

Additionally, besides above characterizations, we have supplemented the Raman spectrum of spent $\text{Al}_2\text{O}_3@\text{ZnS}-\text{S}_d$ to further confirm the formation of $\text{Hg}-\text{S}$ bonds after Hg^0 adsorption. The result showed the, after Hg^0 adsorption, the characteristic peaks of S_d on $\text{Al}_2\text{O}_3@\text{ZnS}-\text{S}_d$ disappeared, and correspondingly, there appeared new peaks located at 249.6 and 343.6 cm^{-1} , which were attributed to the vibration modes of $\text{Hg}-\text{S}$. The related contents are revised as follows:

Revised:

Line 308-312: “Additionally, in the Raman spectrum of spent $\text{Al}_2\text{O}_3@\text{ZnS}-\text{S}_d$, the characteristic peaks of S_d on $\text{Al}_2\text{O}_3@\text{ZnS}-\text{S}_d$ disappeared, and accordingly new peaks located at 249.6 and 343.6 cm^{-1} appeared (Fig. 3b), which were attributed to the vibration modes of $\text{Hg}-\text{S}^{52}$. This signifies that adsorbed Hg^0 can combine with the produced S_n^{2-} to form stable HgS .”

Fig. 3. b Raman spectra of $\text{Al}_2\text{O}_3@\text{ZnS}$, $\text{Al}_2\text{O}_3@\text{ZnS}-\text{S}_d$, spent $\text{Al}_2\text{O}_3@\text{ZnS}-\text{S}_d$, and pure S_8 .

Reference:

52 Minceva-Sukarova, B., Najdoski, M., Grozdanov, I. & Chunnillall, C. J. Raman spectra of thin solid films of some metal sulfides. *J. Mol. Struct.* **410-411**, 267-270, doi:[https://doi.org/10.1016/S0022-2860\(96\)09713-X](https://doi.org/10.1016/S0022-2860(96)09713-X) (1997).

5. After adsorption of Hg^0 , the surface of the material is already HgS crystal, the author said, without desorption of Hg , S_d deposition again, can continue to carry out this cycle process, how to prove that layer by layer while the surface of the adsorbent still has enough space for S_d to be added? Is the S_d activated by the S of HgS or the S of MS , or

is it the S_d itself?

Response: Thanks for your comment. As showed in Fig. 4a, as the SCVD rounds proceeded, the 24-h Hg^0 adsorption breakthrough ratio of $Al_2O_3@ZnS-S_d$ gradually converged to that of $Al_2O_3@HgS-S_d$ (~80%) (Fig. 4a and Fig. 5a); moreover, the adsorption rate of $Al_2O_3@ZnS-S_d$ also gradually converged to that of $Al_2O_3@HgS-S_d$ ($1.05 \text{ mg g}^{-1} \text{ h}^{-1}$) (Fig. 5b).

Fig. 4. a Hg^0 adsorption breakthrough curve of $Al_2O_3@ZnS-S_d$ assisted with intermittent SCVD. Reaction conditions: sorbent mass = 0.4 g, temperature = $120 \text{ }^\circ\text{C}$, $[Hg^0] = (2.5 \pm 0.05) \text{ mg m}^{-3}$, $[SO_2] = 5000 \text{ ppm}$, $[H_2O] = 4\%$, $[H_2S] = 100 \text{ ppm}$ (30 min per 24 h), and total flow rate = 300 mL min^{-1} .

Fig. 5. a Hg^0 adsorption breakthrough curve of $Al_2O_3@HgS-S_d$ assisted with intermittent SCVD. **b** Linear fitting of the adsorption rates of $Al_2O_3@ZnS-S_d$ and $Al_2O_3@HgS-S_d$ at $120 \text{ }^\circ\text{C}$. **c** S 2p XPS spectra of $Al_2O_3@HgS$ and $Al_2O_3@HgS-S_d$. Reaction conditions: sorbent mass = 0.4 g, $[Hg^0] = (2.5 \pm 0.05) \text{ mg/m}^3$, $[SO_2] = 5000$

ppm, [H₂O] = 4%, [H₂S] = 100 ppm (30 min per 24 h), and total flow rate = 300 mL min⁻¹.

Besides the self-sustained adsorption performance, the S and Hg speciation in Al₂O₃@HgS-S_d displayed changes to similar to those of S and Zn in Al₂O₃@ZnS-S_d. Compared to Al₂O₃@HgS, the S 2*p* XPS spectrum of Al₂O₃@HgS-S_d brought out the characteristic peaks of S_n²⁻, which were between the binding energies of S_n²⁻ in Al₂O₃@ZnS-S_d and S⁰ in Al₂O₃-S_d. This indicates that the average chemical valence of deposited S_d on Al₂O₃@HgS was slightly higher than that on Al₂O₃@ZnS, explaining the decrease in Hg⁰ adsorption on Al₂O₃@ZnS with increasing SCVD rounds. Additionally, the Hg 4*f* spectra of Al₂O₃@HgS and Al₂O₃@HgS-S_d were supplemented. The results showed that, similar to Zn 2*p* spectrum in Al₂O₃@ZnS-S_d (Fig. 3d), the locations of Hg²⁺ 4*f* in Al₂O₃@HgS-S_d shifted to lower binding energies (Supplementary Fig. 22).

Moreover, we supplemented the tendency between the adsorption capacity increment (Q_i) of Al₂O₃@HgS-S_d (Supplementary 23a) and the Hg–S bond energy (Supplementary Table 3). The results showed that the corresponding coordinate point likewise fell within the negative correlation trend between the M–S lengths and Q_i of the chalcophile metal sulfides (Supplementary Fig. 23b). Therefore, this confirms the activation effect of HgS itself on S_d, thus enabling the self-sustained adsorption of Hg⁰ on Al₂O₃@MS surface and realizing multilayer adsorption.

On the other hand, although the adsorbent pellet particles increase with the deposition of S_d and adsorption of Hg⁰, the adsorbents can still be used for a long time without affecting the gas air velocity due to their micrometer-level growth rate. According to the EDS mapping images of the cross section of spent Al₂O₃@ZnS-S_d, the thickness of the HgS adsorption layer on the adsorbent is about 50 μm after 10 rounds of deposition-adsorption cycles (Fig. 4b and d). Assuming that the adsorbent can grow by 1 mm, about 200 cycles can be carried out. In addition, Al₂O₃ itself has a porous structure, thus the deposition-adsorption process is not a completely outward growth process.

Fig. 4. b Photos of $\text{Al}_2\text{O}_3@\text{ZnS}$ in different reaction stages. **d** EDS mapping images of the cross section of fresh $\text{Al}_2\text{O}_3@\text{ZnS}$ and spent $\text{Al}_2\text{O}_3@\text{ZnS-S}_d$, respectively. Inset: line scanning results of the selected position.

Therefore, the activation effect of HgS itself and the slow growth rate of HgS layer guaranteed that the of the adsorbent surface has enough space for S_d addition and activated during the self-sustained and multilayer adsorption of Hg^0 .

Revised:

Line 388-398: “The Hg 4*f* spectra showed that, similar to Zn 2*p* spectrum in $\text{Al}_2\text{O}_3@\text{ZnS-S}_d$ (Fig. 3d), the locations of Hg^{2+} 4*f* in $\text{Al}_2\text{O}_3@\text{HgS-S}_d$ shifted to lower binding energies (Supplementary Fig. 22). This indicates that the average chemical valence of deposited unsaturated coordinated S_n^{2-} on $\text{Al}_2\text{O}_3@\text{HgS}$ was slightly higher than that on $\text{Al}_2\text{O}_3@\text{ZnS}$, explaining the decrease in Hg^0 adsorption on $\text{Al}_2\text{O}_3@\text{ZnS}$ with increasing SCVD rounds. Moreover, the relationship between the Q_i of $\text{Al}_2\text{O}_3@\text{HgS-S}_d$ and the Hg–S bond energy likewise fell within the negative correlation trend between the M–S lengths and Q_i of the chalcophile metal sulfides (Supplementary 23, Supplementary Table 3). Therefore, these confirm the activation effect of HgS itself on S_d , thus enabling the self-sustained adsorption of Hg^0 on $\text{Al}_2\text{O}_3@\text{MS}$ surface and realizing multilayer adsorption.”

Supplementary Fig. 22 Hg 4f XPS spectra of Al₂O₃@HgS and Al₂O₃@HgS-S_d.

Supplementary Fig. 23 (a) Hg⁰ adsorption curves of Al₂O₃@HgS and Al₂O₃@HgS-S_d. Reaction conditions: sorbent mass = 0.3 g, SO₂ concentration = 5000 ppm (during SCVD process), H₂S concentration = 100 ppm (during SCVD process), SCVD time = 15 min, Hg⁰ concentration = (1.5±0.05) mg m⁻³, reaction temperature = 80 °C, and total flow rate = 360 mL min⁻¹. (b) The tendency between the adsorption capacity increment and the metal-sulfur bond length of different metal sulfides (including HgS).

Reviewer #2 (Remarks to the Author):

The authors developed a novel and efficient in-situ low-temperature SCVD method using SO₂ and H₂S as feed gas to realize self-sustained adsorption of mercury over metal sulfides sorbents, which is verified via large numbers of experimental tests and characterizations. The presented idea and method is very interesting and enlightening, which can contribute to the development of cost-effective techniques or sorbents for mercury emission control. Thus, a minor revision is recommended.

1. Line 71-73, pertinent literature should be added to support the role of metal sites of MS in activating S_d into S_n²⁻ as this is very important to testify or interpret the major

reaction scheme of the proposed SCVD method.

Response: Thanks for your suggestion. The sentence in Line 71-73 that proposed the role of metal sites of MS in activating S_d into S_n^{2-} is one of the main conclusions of this work. Chalcophile metal element is characterized by a copper-type structure with 18 electrons ($s^2p^6d^{10}$) in the outermost electron layer of its ions, which has high affinity to sulfur. Thus, we choose chalcophile metal sulfides as surface for S_d activation to compare with non-chalcophile metal sulfides. We have added more related literature to support the choice of chalcophile metal sulfides.

Revised:

Line 68-71: “Considering the abundant S^{2-} species in sulfides and the strong affinity between chalcophile elements (whose ion has an 18 outer electron layer structure) and sulfur³⁵⁻³⁷, chalcophile metal sulfides are promising candidates for in-situ SCVD and the species regulation of sulfur.”

Line 77-11: “Results show that metal sites with chalcophile affinity present in sulfides (such as CuS, ZnS, PbS, etc.) can promote the transformation of S_d to active S_n^{2-} species at low temperatures, extremely enhancing the Hg^0 adsorption performance.”

2. Line 77-78 and line 315, the authors claimed that reaction product HgS can also serve as a fresh interface for S_d deposition and activation. Likewise, is there any literature or theory to support this standpoint?

Response: Thanks for your comment. Likewise, the sentence in Line 77-78 is the main findings of our work. We have added the pertinent literature to support the chalcophile nature of Hg element in Line 315.

Revised:

Line 371-372: “..., which may also possess the ability for to activate S_d owing to the chalcophile nature of Hg element⁵³.”

Reference:

53 Frost, B. R., Mavrogenes, J. A. & Tomkins, A. G. PARTIAL MELTING OF SULFIDE ORE DEPOSITS DURING MEDIUM- AND HIGH-GRADE METAMORPHISM. *The Canadian Mineralogist* **40**, 1-18,

doi:10.2113/gscanmin.40.1.1 (2002).

3. Line 146, the sentence "...chalcophile metals have a higher tendency to form highly covalent bonds with sulfur, making them more effective for Hg⁰ adsorption^{35,36}." Several scholars had claimed that the weak strength of the M-S bond is favorable for Hg⁰ adsorption (The sequestration of gaseous Hg⁰ by sphalerite with Fe substitution structure-activity relationship. *J. Phys. Chem. C* 2019, 123, 2828–2836.; Outstanding performance of ZnS/TiO₂ for the urgent disposal of liquid mercury leakage indoors: Novel support effect, reaction mechanism and kinetics. *J. Hazard. Mater.* 403 (2021) 123867), which is contrary to the authors' viewpoint. What is the reason?

Response: Thanks for your comment. The higher tendency of chalcophile metals to form highly covalent bonds with sulfur is compared with other classified elements such as lithophile and siderophile elements, based on Goldschmidt geochemical classification of the elements. For lithophile elements, they tend to form highly ionic bonds with oxygen; while siderophile elements have lower affinity for both sulfur and oxygen. Therefore, the higher affinity of chalcophile metals for sulfur give them higher potential to activate the deposited S_d, thereby leading to the higher Hg⁰ adsorption performance of Al₂O₃@CMS-S_d (CMS, chalcophile metal sulfides). However, the viewpoint that "weak strength of the M-S bond is favorable for Hg⁰ adsorption" is specific to the M-S bond of the metal sulfide itself. To make the point clearer, the sentence has been revised as follows:

Revised:

Line 158-161: "This is due to the higher affinity of chalcophile metals for sulfur, which gives them higher potential to activate the deposited S_d, thereby leading to the higher Hg⁰ adsorption performance of corresponding Al₂O₃@MS-S_d^{36,37}."

4. H₂S is a toxic exhaust gas, thus, the use of H₂S as feed gas is also beneficial in terms of environment protection and waste recycling, this could be mentioned in the introduction section. Moreover, SO₂ is rich in coal-fired flue gas, however, H₂S is very few. Thus, the sources of H₂S should be indicated or discussed.

Response: Thanks for your suggestion. H₂S is one of the important products of fossil fuel combustion, which is a toxic gaseous pollutant. Thus, the use of H₂S as feed gas is beneficial in terms of environment protection and waste recycling. Considering the reviewer's suggestion, we have added the related contents in the Introduction section.

Revised:

Line 52-55: "Meanwhile, hydrogen sulfide (H₂S), as a toxic exhaust gas mainly from fossil fuel combustion²⁴, can be used as feed gas to assist the reaction, which is beneficial in terms of environment protection and waste recycling. Moreover, recently, a fast-induced reduction process was proposed to loop upcycling SO₂ into H₂S²⁵."

References:

- 24 Khan, M. A. H., Rao, M. V. & Li, Q. Recent Advances in Electrochemical Sensors for Detecting Toxic Gases: NO₂, SO₂ and H₂S. *Sensors* **19**, 905 (2019).
- 25 Sun, X. *et al.* Looping upcycling SO₂ into value-added H₂S by fast-induced reduction process for heavy metals treatment in nonferrous smelting industry. *Fuel* **331**, 125867, doi:<https://doi.org/10.1016/j.fuel.2022.125867> (2023).

5. Line 171-172, besides the decomposition of partially formed mercury compounds, is there any other reason for the decrease of Hg⁰ removal performance at 140 °C? Based on Hg-TPD results (Supplementary Fig. 6), it is clear that mercury release only kicks off at 150 °C. Thus, very few formed mercury compounds may decompose at 140 °C. In my opinion, the Claus reaction is significantly affected by reaction temperature. With the increase of temperature, the chain or ring structure of S_x products may also change, possibly leading to decreased Hg⁰ removal performance.

Response: Thanks for your opinion. Our recent research also finds out that increase temperature will induce the ring-opening of S₈ on ZnS surface, and form S_x with different chain lengths, which highly affects its affinity towards Hg⁰. Thus, it is reasonable that the change in S_a speciation will lead to a decrease in Hg⁰ removal performance at high temperatures.

Supplementary Fig. 10 Raman spectra of ZnS and in-situ Raman spectra of ZnS-S_d at temperature range of 25-140 °C.

After considering the reviewer's comments, we have revised the sentence as follows:

Revised:

In manuscript:

Line 195-201: "Fig. 2b and Supplementary Fig. 8 illustrate that increasing in adsorption temperature from 60 to 120 °C improved the Hg⁰ removal efficiency of Al₂O₃@ZnS-S_d from 23.8% to 89.9% (within 180 min). However, when temperature was further elevated to 140 °C and 160 °C, its activity dropped to 86.6% and 75.9%, respectively, presumably due to the re-decomposition of partially captured Hg⁰ (Supplementary Fig. 9) or the decreased Hg⁰ affinity caused by the change in S_d speciation at high temperatures (Supplementary Fig. 10)."

In Supplementary Information:

"To explore the dynamic variation of sulfur species in Al₂O₃@ZnS-S_d. Pure ZnS was synthesized to directly serve as support for SCVD process. The in-situ heating Raman spectra of the resulting ZnS-S_d showed that with temperature increased from 25 to 80 and further to 140 °C, the intensities of the characteristic peaks of S₈ gradually decreased, while the vibration modes of S_x chain increased¹. Yang et al. found that long-chain S_x had negligible adsorption ability, while short-chain S_x had a high affinity for Hg⁰ (ref. ²). As shown, the ratio of short-chain S₃₋₅ (465 cm⁻¹) to long-chain S₆₋₈ (414

cm⁻¹) decreased from 1.40 at 80 °C to 0.93 at 140 °C³. This could also explain the decrease in Hg⁰ adsorption activity at temperatures above 120 °C.”

6. Line 196-197, the reasons for the positive effect of SO₂ on Hg⁰ removal could be discussed more.

Response: Thanks for your suggestion. The reason for the positive effect of SO₂ on Hg⁰ removal over Al₂O₃@ZnS-S_d is probable due to the formation of nascent active sites. Through Hg⁰ temperature-programmed desorption (Hg⁰-TPD) experiments, we found that the desorption peaks of spent Al₂O₃@ZnS-S_d in the absence of SO₂ mainly located at 210 and 265 °C, which are attributed to the decomposition temperatures of β-HgS and α-HgS, respectively. After adsorption in the presence of SO₂, the proportion of β-HgS decreased and there appeared new peak centered at 365 °C, which can be assigned to HgO. Thus, it can be speculated that, during the adsorption process, SO₂ may react with the newly created weaker structured β-HgS to form HgO, and simultaneously generate new S_d sites (SO₂ + 2HgS → 3S_d + 2HgO). The related contents are revised as follows:

Revised:

Line 229-239: “Through Hg⁰ temperature-programmed desorption (Hg⁰-TPD) experiments, we analyzed the changes in de-mercury products (Supplementary Fig. 9). The desorption peaks of spent Al₂O₃@ZnS-S_d in the absence of SO₂ mainly located at 210 °C (63.0%) and 265 °C (37%), which are attributed to the decomposition temperatures of β-HgS and α-HgS, respectively^{42,43}. While, after adsorption in the presence of SO₂, the proportion of β-HgS decreased to 32.4%, and there appeared new peak centered at 365 °C (33.1%), which is assigned to the decomposition temperature of HgO⁴⁴. Thus, it can be speculated that, during the adsorption process, SO₂ may react with the weaker structured β-HgS to form more stable HgO, and simultaneously produce nascent S_d sites (SO₂ + 2HgS → 3S_d + 2HgO), thereby improving its Hg⁰ adsorption performance.”

References:

42 Rumayor, M., Diaz-Somoano, M., Lopez-Anton, M. A. & Martinez-Tarazona,

M. R. Mercury compounds characterization by thermal desorption. *Talanta* **114**, 318-322, doi:<https://doi.org/10.1016/j.talanta.2013.05.059> (2013).

- 43 Hong, Q. *et al.* Stepwise Ions Incorporation Method for Continuously Activating PbS to Recover Mercury from Hg⁰-Rich Flue Gas. *Environ. Sci. Technol.* **54**, 11594–11601, doi:10.1021/acs.est.0c03335 (2020).
- 44 Xu, H. *et al.* Stabilization of mercury over Mn-based oxides: Speciation and reactivity by temperature programmed desorption analysis. *J. Hazard. Mater.* **321**, 745-752, doi:10.1016/j.jhazmat.2016.09.030 (2017).

Supplementary Fig. 9 Hg⁰ Temperature programmed desorption (Hg⁰-TPD) curves of spent Al₂O₃@ZnS-S_d after Hg⁰ adsorption with or without the presence of SO₂.

7. Line 237, the peak at 164.6 is probably owing to elemental sulfur S₈, as confirmed by Raman spectra in Fig. 3b, S₈ is also present on Al₂O₃@ZnS-S_d.

Response: Thanks for your comment. The difference in binding energy between the two peaks at 163.4 and 164.6 eV is 1.2 eV with an intensity ratio of 2:1, thus they are more likely to be attributed to the 2p_{3/2} and 2p_{1/2} splitting peaks of S_n²⁻, respectively. The Raman spectrum of Al₂O₃@ZnS-S_d exhibited the vibration modes of S-S bonds, but their Raman shifts were higher than those of elemental sulfur S₈, indicating the non-physical combination of the deposited S_d on Al₂O₃@ZnS. To make a clearer expression

of this sentence, it has been revised as follows:

Revised:

Line 277-280: “For Al₂O₃@ZnS-S_d, aside from the peaks ascribed to S²⁻ and SO₄²⁻, two new peaks located at 163.4 and 164.6 eV occurred, which can be assigned to the 2p_{3/2} and 2p_{1/2} splitting peaks respectively, of S_n²⁻ sites formed by S_d deposited on Al₂O₃@ZnS (middle layer in Fig. 3c)⁴¹.”

8. How to prepare Al₂O₃@HgS sorbent? Please add this information in Methods section as well.

Response: Thanks for your suggestion. We have added the synthesis method of Al₂O₃@HgS in the Methods section. The related contents are revised as follows:

Revised:

Line 441-455: “Different metal sulfides (MS, M = Cu, Zn, Cd, In, Hg, Pb, Sn Mn, Fe, Co, and Ni) were synthesized via a one-step hydrothermal method as the interface for SCVD and Hg⁰ adsorption. ... Other Al₂O₃@MS adsorbents were synthesized in the same way, except that different metal precursors, including CuSO₄·5H₂O (2.5 mmol), CdCl₂ (2.5 mmol), InCl₃·4H₂O (1.67 mmol), HgCl₂ (2.5 mmol), (CH₃COO)₂Pb (2.5 mmol), SnCl₂ (2.5 mmol), MnSO₄·H₂O (2.5 mmol), FeCl₃ (1.67 mmol), CoCl₂·6H₂O (2.5 mmol), and NiCl₂·6H₂O (2.5 mmol), were used instead of ZnSO₄·7H₂O.”

9. Line 176-177, the reasons for the negative effect of H₂O on mercury adsorption should be discussed more. As reported, steam shows a complicated influence on Hg⁰ capture over metal sulfides or sulfur-bearing materials (Novel metal sulfide sorbents for elemental mercury capture in flue gas: A review. Fuel DOI: 10.1016/j.fuel.2023.129829), insignificant, promotional, suppressive effects had all been reported.

Response: Thanks for your suggestion. H₂O is an important component in flue gases, and thus it is necessary to consider its effect on the performance of adsorbents. The negative effect of H₂O on the Hg⁰ adsorption performance over Al₂O₃@ZnS-S_d is probably ascribed to the competing adsorption between H₂O and Hg⁰ on the active sites

and the hydrophilicity of Al₂O₃.

Revised:

Line 206-210: “The introduction of 4% H₂O showed negative effect on Hg⁰ adsorption performance of Al₂O₃@ZnS-S_d, presumably due to the competing adsorption between H₂O and Hg⁰ on the active sites⁴⁰ and the hydrophilicity of Al₂O₃⁴¹; however, its removal efficiency maintained at a stable level of ~70% without reduction for 180 min adsorption (Supplementary Fig. 11).”

References:

- 40 Liu, D., Li, C., Jia, T., Wu, J. & Li, B. Novel metal sulfide sorbents for elemental mercury capture in flue gas: A review. *Fuel* **357**, 129829, doi:https://doi.org/10.1016/j.fuel.2023.129829 (2024).
- 41 Li, Z. *et al.* Preparation of Al₂O₃-coated expanded graphite with enhanced hydrophilicity and oxidation resistance. *Ceram. Int.* **44**, 16256-16264, doi:https://doi.org/10.1016/j.ceramint.2018.06.017 (2018).

10. Supplementary Table 4, the breakthrough percent for each sorbent could be provided for facile comparison. Line 24 and 359, the breakthrough percent should be specified as well.

Response: Thanks for your suggestion. We have added the breakthrough ratio of each sorbent in Supplementary Table 4 (Revised Supplementary Table 6). However, for self-sustained Al₂O₃@ZnS-S_d adsorbent, its Hg⁰ adsorption activity can be continuously restored via intermittent SCVD method (as shown in Fig. 4a), thus it did not have a specific breakthrough ratio. In fact, as progressing with the SCVD rounds, the 24-h Hg⁰ adsorption breakthrough ratio of Al₂O₃@ZnS-S_d gradually converged to ~80%, which is close to that of Al₂O₃@HgS-S_d.

Revised:

Line 408-410: “Fig. 6 and Supplementary Table 6 compare the Hg⁰ adsorption capacities of Al₂O₃@ZnS-S_d with various reported adsorbents, including carbon-based, oxides-based, and sulfide-based materials.”

Line 418-421: “Impressively, the self-sustained Al₂O₃@ZnS-S_d not only reversed the

poisoning effect of SO₂ but also reached a Hg⁰ adsorption capacity of 303.92 mg g⁻¹ (normalized to ZnS coating amount, and with 24-h breakthrough ratio of ~80%) after 236 h reaction, ...”

Supplementary Table 6 Comparison of adsorption capacity of different kinds of mercury sorbents.

	Sorbents	Gas component	Adsorption capacity (mg/g)	Ref.
Carbon-based	ACN-AR	N ₂	0.236 (100%) ^a	6
	Coal (FGD)	6%O ₂ +12%CO ₂ +7%H ₂ O	0.573 (100%)	7
	Biomass (PAC)	6%O ₂ +12%CO ₂ +7%H ₂ O	0.383 (100%)	
	Fly ash carbon	16%CO ₂ +5%O ₂ +2000ppmSO ₂	1.85 [350 min] ^b	8
	S/AC	N ₂	2.3 (100%)	9
	Cu-BTC	10%O ₂ (15 ppm HCl for 5 min)	1.7 [2 h]	10
Oxide-based	α-MnO ₂	4%O ₂	~6.94 [10 h]	11
	15%-Mn/γ-Fe ₂ O ₃ -250	air	3.54 (55%)	12
	MnO ₂ /CS	N ₂	~4.78 [10 h]	13
	(Fe ₂ Ti)0.8O ₄	air	3.94 (23%)	14
	MnO _x /graphene-30%	4% O ₂	2.7 [10 h]	15
	Ce _{0.5} Mn _{0.5} O _y	4%O ₂ +500ppmNO+500ppm SO ₂	5.6 [10 h]	16
	(Fe _{2.2} Mn _{0.8}) _{1-δ} O ₄	10%O ₂	4.44 [10 h]	17
	Fe-Sn-MnO _x (1:20:20)	4%O ₂	~3.75 [10 h]	18
	Fe-Ti-Mn spinel	4%O ₂	2.3 [10 h]	19
	LaMnO ₃	8%O ₂	6.8 [10 h]	20
	MnO _x /CeO ₂ -TiO ₂	400ppmNO+400ppmCO	9.4 (100%)	21
Sulfur-based	Nano-CuS	N ₂	122.4 (100%)	22
	Fe ₃ O ₄ @CuS	N ₂	88.7 (100%)	23
	In-situ etching ZnS	N ₂	53.83 (50%)	4
	FeS _{1.32} Se _{0.11}	6%O ₂ +2.5%SO ₂ +5%H ₂ O	20.216 (100%)	24
	NiAl-S ₄ @SiO ₂ -urchin	4%O ₂	7.65 [5 h]	25
	[MoS ₄] ²⁻ /CoFe-LDH	4%O ₂	16.39 (100%)	26
	MoS ₃ /TiO ₂	N ₂	14.9 (75%)	27
	CoS _x	4%O ₂	43.03 (50%)	28
	CoMoS/γ-Al ₂ O ₃	N ₂	18.94 (100%)	29
	In ₂ S ₃ /g-C ₃ N ₄	N ₂	14.78 (100%)	30
	CuInS ₂	N ₂	13.81 (100%)	31
	40ZIS/CN	N ₂	13.04 (100%)	32
	CuS/Al ₂ O ₃	N ₂	20.96 (100%)	33
	Mn-Sn ₂ S ₆	N ₂	21.05 (100%)	34
	Cu ₂ S	N ₂	26.6 (100%)	35
	CuFeS ₂	N ₂	37.24 (100%)	36
	FeMoS _x /TiO ₂	N ₂	41.8 (100%)	37
Co ₉ S ₈	N ₂	43.18 (100%)	38	

MoS ₂	N ₂	54.308 (100%)	39
ZnO@CuS	N ₂	60.53 (100%)	40
Al ₂ O ₃ @ZnS-S _d	5000ppmSO ₂ +4%H ₂ O +100ppmH ₂ S(30min/24h)	303.92	This work

a: Adsorption breakthrough ratio; b: Adsorption time.

11. Fig 1b and Supplementary Fig. 3, 7, 13, and 15, it is suggested to add the reaction conditions (gas composition, reaction temperature, sorbent dosage, adsorption time) in the figure captions to make the figures more readable. Specially, the breakthrough percents could be offered in Fig. 1b. Because Fig. 1b is the crucial data to testify the idea of this work and would be valuable for peers to refer to.

Response: Thanks for your suggestion. We have added the reaction conditions in Fig. 1b and Supplementary Fig. 3, 7, 13, and 15. For the breakthrough ratio of different adsorbents in Fig. 1b, we have added a table in Supplementary Information. The related contents are revised as follow:

Revised:

Fig. 1b: “Hg⁰ adsorption capacities of different Al₂O₃@MS and Al₂O₃@MS-S_d (M = Cu, Zn, Cd, In, Pb, Sn, Ni, Co, Fe, and Mn). Reaction conditions: sorbent mass = 0.3 g, total flow rate = 360 mL min⁻¹, SO₂ concentration = 5000 ppm (during SCVD process), H₂S concentration = 100 ppm (during SCVD process), SCVD time = 15 min, Hg⁰ concentration = (1.5±0.05) mg m⁻³, reaction temperature = 80 °C, and reaction time = 180 min.”

Supplementary Fig. 3: “**Supplementary Fig. 3** Effect of the addition sequence of H₂S and SO₂ on the Hg⁰ adsorption performance of Al₂O₃@ZnS-S_d. Reaction conditions: sorbent mass = 0.3 g, total flow rate = 360 mL min⁻¹, SCVD time = 15 min, Hg⁰ concentration = (1.5±0.05) mg m⁻³, reaction temperature = 80 °C.”

Supplementary Fig. 7: “**Supplementary Fig. 11** Effect of different gas components on Hg⁰ adsorption performance over Al₂O₃@ZnS-S_d. Reaction conditions: sorbent mass = 0.3 g, SO₂ concentration = 5000 ppm (during SCVD process), H₂S concentration = 100 ppm (during SCVD process), SCVD time = 15 min, Hg⁰ concentration = (1.5±0.05) mg m⁻³, reaction temperature = 120 °C, and total flow rate = 360 mL min⁻¹.”

Supplementary Fig. 13: “**Supplementary Fig. 19** Temperature effect on the Hg⁰ removal efficiency of Al₂O₃@ZnS-S_d and Al₂O₃@CuS-S_d. Reaction conditions: sorbent mass = 0.3 g, SO₂ concentration = 5000 ppm (during SCVD process), H₂S concentration = 100 ppm (during SCVD process), SCVD time = 15 min, Hg⁰ concentration = (1.5±0.05) mg m⁻³, and total flow rate = 360 mL min⁻¹.”

Supplementary Fig. 15: “**Supplementary Fig. 24** Hg⁰ adsorption performance of in-situ etching ZnS and Al₂O₃@ZnS. Reaction conditions: sorbent mass = 20 mg of in-situ etching ZnS or 0.3 g of in-situ etching Al₂O₃@ZnS, reaction temperature = 120 °C, and total flow rate = 360 mL min⁻¹.”

Line 144-147: “As depicted in Fig. 1b, after 15 min of SCVD, various Al₂O₃@MS with S_d deposition (Al₂O₃@MS-S_d) showed substantial differences in enhancing their Hg⁰ adsorption capacities (within 180 min and normalized to sulfide molar mass), and Supplementary Table 2 summarized their Hg⁰ adsorption breakthrough ratio during the reaction process.”

Supplementary Table 2 The Hg⁰ adsorption breakthrough ratio of different Al₂O₃@MS and Al₂O₃@MS-S_d.

Adsorbents	Breakthrough ratio (0~180 min)	Adsorbents	Breakthrough ratio (0~180 min)
Al ₂ O ₃ @CuS	8.1%~90.2%	Al ₂ O ₃ @CuS-S _d	3.0%~8.1%
Al ₂ O ₃ @ZnS	87.9%~99.4%	Al ₂ O ₃ @ZnS-S _d	20.6%~36.1%
Al ₂ O ₃ @CdS	79.0%~93.6%	Al ₂ O ₃ @CdS-S _d	9.9%~30.8%
Al ₂ O ₃ @In ₂ S ₃	82.0%~93.6%	Al ₂ O ₃ @In ₂ S ₃ -S _d	25.0%~42.8%
Al ₂ O ₃ @PbS	58.6%~84.4%	Al ₂ O ₃ @PbS-S _d	23.1%~44.4%
Al ₂ O ₃ @SnS	69.1%~92.5%	Al ₂ O ₃ @SnS-S _d	39.1%~62.5%
Al ₂ O ₃ @Fe ₂ S ₃	51.1%~93.5%	Al ₂ O ₃ @Fe ₂ S ₃ -S _d	66.3%~67.5%
Al ₂ O ₃ @CoS	48.6%~88.6%	Al ₂ O ₃ @CoS-S _d	65.5%~78.6%
Al ₂ O ₃ @MnS	70.0%~91.6%	Al ₂ O ₃ @MnS-S _d	55.9%~83.5%
Al ₂ O ₃ @NiS	81.9%~100%	Al ₂ O ₃ @NiS-S _d	71.7%~88.1%

* Reaction conditions: sorbent mass = 0.3 g, total flow rate = 360 mL min⁻¹, SO₂ concentration = 5000 ppm (during SCVD process), H₂S concentration = 100 ppm (during SCVD process), SCVD time = 15 min, Hg⁰ concentration = (1.5±0.05) mg m⁻³, reaction temperature = 80 °C, and reaction time = 180 min.

12. In Scheme 1, some texts are needed to briefly describe the experimental procedure of SCVD method in the figure caption.

Response: Thanks for your suggestion. We have added a brief description of experimental procedure of SCVD method in the caption of Scheme 1.

Revised:

Scheme 1. Schematic illustration of the proposed in-situ SCVD technology for self-sustained and multilayer adsorption of Hg^0 . Chemical vapor deposition of sulfur is achieved by the reaction of H_2S with SO_2 adsorbed on the $\text{Al}_2\text{O}_3@\text{MS}$ and activated to S_n^{2-} species by surface metal sites; when S_n^{2-} is occupied by Hg^0 , it can be replenished by repeated SCVD, resulting in self-sustained and multilayer adsorption of Hg^0 .

13. In our experimental experience, H_2S could significant disable the Hg^0 detector, such as VM3000. Is it also for Lumex? How did the authors deal with this issue?

Response: Thanks for your comment. As shown in the following figures, we tested the effect of H_2S on the signal of Hg^0 concentration recorded by Lumex RA915+. The results showed that the addition of H_2S did not affect the baseline concentration of Hg^0 with or without the presence of SO_2 . However, when the inlet Hg^0 concentration increased to about $1500 \mu\text{g m}^{-3}$, the addition of 100 ppm H_2S resulted in a decrease in Hg concentration to about $1400 \mu\text{g m}^{-3}$ (Figure R2a); while, in the presence of 5000 ppm SO_2 , switching on or off 100 ppm H_2S did not affect the Hg^0 signal (Figure R2b). Moreover, as shown in the Supplementary Fig. 2a, the reaction of 200 ppm SO_2 with 100 ppm H_2S reduced the H_2S concentration to about 12 ppm. Thus, although that H_2S alone can reduce the Hg^0 signal detected by the Lumex RA915+ by about 6.7%, its effect can be eliminated by coexistence with SO_2 due to their mutual reaction. In this work, H_2S was injected to the stream for 30 min per 24 h in the presence of SO_2 to

perform the SCVD reaction, so the effect of H₂S on the Hg⁰ signal can be ignored.

Figure R2. Effect of H₂S on the Hg⁰ signal detected by Lumex RA915+: (a) without SO₂ and (b) with SO₂. H₂S concentration = 100 ppm, SO₂ concentration = 5000 ppm, temperature = 120 °C, total flow rate = 360 mL min⁻¹

14. Line 66, the sentence ‘...and the species tuning of sulfur...’ is confusing, please rephrase it to make the idea clearer.

Response: Thanks for your comment. We are sorry for the unclear expression of the sentence and we have revised this sentence as follow.

Revised:

Line 70-71: “... chalcophile metal sulfides are promising candidates for in-situ SCVD and the species regulation of sulfur.”

15. Some typos:

--Line 68, a word ‘over or on’ may be missing prior to ‘...metal sulfides’.

--Fig. 3, the figure caption should be corrected, which seems incorrect.

--Line 323-324, ‘...adsorption performance of with Al₂O₃@HgS-S_d at 60 °C.’

Response: Thanks for your careful review. We have corrected all typos and double-checked the entire manuscript.

Revised:

Line 72-74: “Therefore, we propose an innovative in-situ SCVD method to counteract the negative effects of SO₂ and achieve self-sustained Hg⁰ adsorption on metal sulfides coated γ-Al₂O₃ pellets (Al₂O₃@MS)”

Fig. 3: “**a** XRD patterns of $\text{Al}_2\text{O}_3@\text{ZnS}$, $\text{Al}_2\text{O}_3@\text{ZnS-S}_d$, and spent $\text{Al}_2\text{O}_3@\text{ZnS-S}_d$. **b** Raman spectra of $\text{Al}_2\text{O}_3@\text{ZnS}$, $\text{Al}_2\text{O}_3@\text{ZnS-S}_d$, spent $\text{Al}_2\text{O}_3@\text{ZnS-S}_d$, and pure S_8 . **c** S $2p$ and **d** Zn $2p$ XPS spectra of $\text{Al}_2\text{O}_3@\text{ZnS}$, $\text{Al}_2\text{O}_3@\text{ZnS-S}_d$, and spent $\text{Al}_2\text{O}_3@\text{ZnS-S}_d$.”

Line 380-381: “... we further compared its self-sustained adsorption performance with $\text{Al}_2\text{O}_3@\text{HgS-S}_d$ at 60 °C.”

Reviewer #3 (Remarks to the Author):

The removal of flue gas mercury using adsorbents is vital, but current adsorbents face challenges due to complex synthesis, poor resistance to SO_2 , and low adsorption capacities. Hong et al. observed an intriguing phenomenon that flue gas SO_2 can directly transform into active sulfur sites on metal sulfide surfaces. Based on this discovery, they proposed an innovative in-situ sulfur CVD method for flue gas mercury capture, ensuring sustained self-sustained mercury adsorption performance. This approach continuously converts harmful flue gas SO_2 into nascent sulfur species and effectively utilizes chalcophile metals to modulate sulfur speciation, achieving prolonged high-capacity mercury adsorption. Therefore, I think it is suitable to be published in Nature Communications after a minor revision. The related comments are as follows:

1) In the Abstract section, the full name of S_n^{2-} should be given, and there is more than one expression for S_n^{2-} throughout the manuscript (line 54, line 72), please standardize the expression to avoid confusion.

Response: Thanks for your suggestion. The full name of S_n^{2-} in the Abstract has been added and we have standardized the expression of S_n^{2-} throughout the revised manuscript.

Revised:

Line 20-21: “The affordance of MS to activate S_d to polysulfide (S_n^{2-}) species depends on the sulfur affinity of its metal element.”

Line 59-61: “Among various sulfur species, the polysulfide (S_n^{2-}) is identified as an

important active species due to the strong oxidation and affinity properties of its terminal sulfur towards Hg^0 (ref. ^{26,27}).”

Line 77-79: “Metal sites with chalcophile affinity present in sulfides (such as CuS, ZnS, PbS etc.) can promote the transformation of S_d to active S_n^{2-} species at low temperatures, ...”

2) Line 68, please give a clear definition of “self-sustained”. What is the relationship between “self-sustained” and “multilayer” adsorption?

Response: Thanks for your comment and we are sorry for the unclear definition of the sentence. The word “self-sustained” in our manuscript means that the spent adsorbents does not need to be replaced but can be re-activated by the proposed SCVD technology. Through intermittent SCVD (30 min per 24 h) and Hg^0 adsorption, namely the self-sustained adsorption of Hg^0 , the S_d and Hg^0 can continuously deposited or adsorbed on $\text{Al}_2\text{O}_3@\text{ZnS}$ surface layer by layer, so as to realize the multilayer adsorption of Hg^0 . To make it easier to understand, we have added the definition of “self-sustained” and clarify its relationship with “multilayer” adsorption.

Revised:

Line 72-81: “Therefore, we propose an innovative in-situ SCVD method to counteract the negative effects of SO_2 and achieve self-sustained Hg^0 adsorption on metal sulfides coated $\gamma\text{-Al}_2\text{O}_3$ pellets ($\text{Al}_2\text{O}_3@\text{MS}$), that is, enabling the in-situ reactivation without replacing the spent adsorbents. (Scheme 1). ... By employing intermittent SCVD, $\text{Al}_2\text{O}_3@\text{ZnS-S}_d$, for example, can achieve self-sustained adsorption of Hg^0 and thus multilayer adsorption.”

3) The XPS spectra of Zn 2p should be added in the Supplementary Information.

Response: Thanks for your suggestion. We have added the Zn 2p XPS spectra of $\text{Al}_2\text{O}_3@\text{ZnS}$, $\text{Al}_2\text{O}_3@\text{ZnS-S}_d$, and spent $\text{Al}_2\text{O}_3@\text{ZnS-S}_d$ in the Supplementary Information. The results showed that the Zn 2p spectra in $\text{Al}_2\text{O}_3@\text{ZnS}$ shifted to lower binding energy after S_d deposition. The related analysis was added in the revised manuscript as follows:

Revised:

Line 291-294: “Moreover, the deposition of S_d also influenced the speciation of Zn element in Al₂O₃@ZnS. The binding energy of Zn²⁺ 2p_{3/2} in Al₂O₃@ZnS shifted from 1022.0 to 1021.8 eV after S_d deposition (Fig. 3d), indicating the formation of unsaturated coordination environments⁵¹.”

Fig. 3 d Zn 2p XPS spectra of Al₂O₃@ZnS, Al₂O₃@ZnS-S_d, and spent Al₂O₃@ZnS-S_d.

4) Considering the application temperature window of the adsorbent, additional experiments are recommended to evaluate the thermal stability and the efficiencies of the adsorbent.

Response: Thanks for your suggestion. The thermal stability of the adsorbents is an important factor affecting their performance. As shown in Fig. 2b, we have supplemented the performance of Al₂O₃@ZnS-S_d at temperatures of 140 and 160 °C. The results showed that the Hg⁰ removal efficiency of Al₂O₃@ZnS-S_d decreased to 86.6% (140 °C) and 75.9% (160 °C) within 180 min. This is presumably due to the re-decomposition of partially captured Hg⁰ or the decreased Hg⁰ affinity caused by the change in S_d speciation at high temperatures. Besides the effect of temperature on the Hg⁰ adsorption performance of Al₂O₃@ZnS-S_d, we also supplemented its thermal stability and compared it with Al₂O₃ and Al₂O₃@ZnS. The results showed that the mass loss of Al₂O₃@ZnS-S_d at 200 °C was reduced by 2.6% and 1.3% compared with that of Al₂O₃ and Al₂O₃@ZnS, respectively (Supplementary Fig. 7), indicating the high thermal stability of ZnS and S_d deposited on Al₂O₃@ZnS. Moreover, the SCVD strategy

is mainly designed for the Hg^0 removal from flue gas during or after the dynamic wave scrubber, where the temperature range is between 60 and 100 °C, in non-ferrous smelting process. Thus, $\text{Al}_2\text{O}_3@\text{ZnS-S}_d$ can be applied to this temperature range. The related supplemented contents are revised in the manuscript as follows:

Revised:

Line 192-195: “Thermogravimetric analysis (TGA) results showed that the mass loss of $\text{Al}_2\text{O}_3@\text{ZnS-S}_d$ at 200 °C was reduced by 2.6% and 1.3% compared with that of Al_2O_3 and $\text{Al}_2\text{O}_3@\text{ZnS}$, respectively (Supplementary Fig. 7), reflecting the high thermal stability of ZnS and S_d deposited on $\text{Al}_2\text{O}_3@\text{ZnS}$.”

Supplementary Fig. 7 Thermo gravimetric analysis of Al_2O_3 , $\text{Al}_2\text{O}_3@\text{ZnS}$, and $\text{Al}_2\text{O}_3@\text{ZnS-S}_d$.

Line 195-201: “Fig. 2b and Supplementary Fig. 8 illustrate that increasing in adsorption temperature from 60 to 120 °C improved the Hg^0 removal efficiency of $\text{Al}_2\text{O}_3@\text{ZnS-S}_d$ from 23.8% to 89.9% (within 180 min). However, when temperature was further elevated to 140 °C and 160 °C, its activity dropped to 86.6% and 75.9%, respectively, presumably due to the re-decomposition of partially captured Hg^0 (Supplementary Fig. 9) or the decreased Hg^0 affinity caused by the change in S_d speciation at high temperatures (Supplementary Fig. 10).”

Fig. 2. b The effects temperature and gas components on Hg⁰ removal efficiency of activated Al₂O₃@ZnS.

Supplementary Fig. 8 Hg⁰ adsorption curves of Al₂O₃@ZnS-S_d at different temperatures. Reaction conditions: sorbent mass = 0.3 g, SO₂ concentration = 5000 ppm (during SCVD process), H₂S concentration = 100 ppm (during SCVD process), SCVD time = 15 min, Hg⁰ concentration = (1.5±0.05) mg m⁻³, and total flow rate = 360 mL min⁻¹.

5) Figure 4a shows great sustainable Hg⁰ adsorption performance of Al₂O₃@ZnS-S_d after each round of SCVD. I think it is necessary to further investigate the Hg⁰ re-emission after multiple rounds of experiments.

Response: Thanks for your suggestion. The issue of Hg⁰ re-emission from spent adsorbents can pose a serious environmental risk, thus this is really an important factor that should be considered. Therefore, we further supplemented the Hg⁰ re-emission experiments of the spent adsorbent after several rounds of SCVD and Hg⁰ adsorption. The results showed that after cutting off the injection of Hg⁰, at the reaction temperature

of $\text{Al}_2\text{O}_3@\text{ZnS-S}_d$ ($120\text{ }^\circ\text{C}$), the Hg^0 concentration reduced to $177\text{ }\mu\text{g}/\text{m}^3$ and $116\text{ }\mu\text{g}/\text{m}^3$ under N_2 and $\text{SO}_2+\text{H}_2\text{O}$ conditions, respectively, within a 60 min purge. While, with the addition of 100 ppm of H_2S , the Hg^0 concentration can decrease to lower than $50\text{ }\mu\text{g}/\text{m}^3$ (emission standard for non-ferrous smelting flue gas in China) in in 30 min, and once the temperature dropped to room temperature, the Hg^0 concentration rapidly decreased to 0. For $\text{Al}_2\text{O}_3@\text{CuS-S}_d$ reacted at $60\text{ }^\circ\text{C}$, the Hg^0 concentration reduced to $50\text{ }\mu\text{g}/\text{m}^3$ within 18, 10, and 3 min for N_2 , $\text{SO}_2+\text{H}_2\text{O}$, and $\text{SO}_2+\text{H}_2\text{S}+\text{H}_2\text{O}$ purging conditions, respectively. The related contents were revised in the manuscript and Supplementary Information as follows:

Revised:

In manuscript:

Line 334-341: “Additionally, we evaluated the Hg^0 re-emission of $\text{Al}_2\text{O}_3@\text{ZnS-S}_d$ after 10 rounds of SCVD and adsorption. At the reaction temperature, the Hg^0 re-emission concentration can be reduced from about $1.7\text{ mg}/\text{m}^3$ to $0.05\text{ mg}/\text{m}^3$ (emission standard for non-ferrous smelting flue gas in China) in 30 min with the assistance of SCVD, and once the temperature dropped to room temperature, the Hg^0 concentration rapidly decreased to 0 (Supplementary Fig. 16). Thus, the SCVD strategy not only can fulfill the self-sustained adsorption of Hg^0 , but also inhibit the re-emission of adsorbed mercury.”

In Supplementary Information:

Supplementary Fig. 16 Hg^0 re-emission experiments of spent (a) $\text{Al}_2\text{O}_3@\text{ZnS-S}_d$ and (b) $\text{Al}_2\text{O}_3@\text{CuS-S}_d$. Reaction conditions: sorbent mass = 0.4 g, $[\text{Hg}^0] = 0\text{ mg m}^{-3}$, $[\text{SO}_2] = 5000\text{ ppm}$ (when used), $[\text{H}_2\text{O}] = 4\%$ (when used), $[\text{H}_2\text{S}] = 100\text{ ppm}$ (when used), and

total flow rate = 300 mL min⁻¹.

“The results showed that after cutting off the injection of Hg⁰, at the reaction temperature of Al₂O₃@ZnS-S_d (120 °C), the Hg⁰ concentration reduced to 177 µg/m³ and 116 µg/m³ under N₂ and SO₂+H₂O conditions, respectively, within a 60 min purge (Supplementary Fig. 16a). While, with the addition of 100 ppm of H₂S, the Hg⁰ concentration can decrease to lower than 0.05 mg/m³ (emission standard for non-ferrous smelting flue gas in China) in 30 min, and once the temperature dropped to room temperature, the Hg⁰ concentration rapidly decreased to 0 (Supplementary Fig. 16a). For Al₂O₃@CuS-S_d reacted at 60 °C, the Hg⁰ concentration reduced to 50 µg/m³ within 18, 10, and 3 min for N₂, SO₂+H₂O, and SO₂+H₂S+H₂O purging conditions, respectively (Supplementary Fig. 16b).”

6) In Lines 324-326, besides the adsorption rates of Al₂O₃@CuS-S_d, the corresponding self-sustained adsorption curve should be provided in the Supplementary information as well.

Response: Thanks for your suggestion. The self-sustained adsorption curve of Al₂O₃@CuS-S_d at 60 °C was added in the revised Supplementary Information.

Revised:

Supplementary Fig. 20 Self-sustained Hg⁰ adsorption curve of Al₂O₃@CuS-S_d at 60 °C. Reaction conditions: sorbent mass = 0.4 g, temperature = 60 °C, [Hg⁰] = (2.5±0.05) mg m⁻³, [SO₂] = 5000 ppm, [H₂O] = 4%, [H₂S] = 100 ppm (30 min 24 h⁻¹), and total flow rate = 300 mL min⁻¹.

“As present in Supplementary Fig. 20, after depleting the initial activity of the Al₂O₃@CuS, 30 min of in-situ SCVD restored its Hg⁰ removal efficiency to ~97%.

After 6 rounds of SCVD, its Hg^0 removal efficiency still reached ~87%, and its 24-h breakthrough ratio was converged to ~50%.”

7) The synthetic method of in-situ etching ZnS and $\text{Al}_2\text{O}_3@\text{ZnS}$ in Supplementary Fig. 15 should be added.

Response: Thanks for your suggestion. The in-situ etching ZnS was synthesized according to the published paper (Li et al., 2023), and the in-situ etching $\text{Al}_2\text{O}_3@\text{ZnS}$ was further prepared by introducing 1-2 mm Al_2O_3 pellets during the synthesise process. The detailed experimental methods were supplemented in the revised Supplementary Information:

Revised:

In Supplementary Information:

“The in-situ etching ZnS was synthesized according to the published paper¹. Typically, 0.01 mol of $\text{ZnSO}_4 \cdot 7\text{H}_2\text{O}$ was dissolved in 50 mL of acid solution (0.368 mol L^{-1} of H_2SO_4), and 0.01 mol of $\text{Na}_2\text{S} \cdot 9\text{H}_2\text{O}$ was dissolved in 50 mL of deionized water separately. Then, the Na_2S solution was added into the ZnSO_4 solution to obtain a light-yellow precipitate. The precipitate was separated by centrifugation and washed several times with deionized water and ethanol. Finally, the precipitate was dried in vacuum at 60°C for 12 h to obtain in-situ etching ZnS.

The in-situ etching $\text{Al}_2\text{O}_3@\text{ZnS}$ was synthesized by a similar method. The difference is that 50 mL of ZnSO_4 solution was first mixed with 20 g of 1–2 mm Al_2O_3 pellets and ultrasonicated for 30 min and then dried to obtain $\text{Al}_2\text{O}_3@\text{ZnSO}_4$.”

8) The paragraph in Lines 364-373 only discusses the applicability of $\text{Al}_2\text{O}_3@\text{MS-S}_d$ in smelting flue gas, while in the Abstract, it is mentioned that it “enables efficient on-site mercury removal in various scenarios”. Please add the discussion of its applicability in other scenarios or revise the statement in the Abstract.

Response: Thanks for your suggestion. SO_2 and H_2O are typical components in most industrial flue gas, like coal-fired flue gas, incineration flue gas, cement flue gas, and non-ferrous smelting flue gas. Among them, non-ferrous metal smelting flue gas is

characterized by high Hg⁰, high SO₂, and high H₂O concentrations, which calls for highly efficient Hg⁰ removal technology. Therefore, smelting flue gas can be a representative application scenario for the SCVD method. We have revised this paragraph to make a clearer expression of its application scenarios.

Revised:

Line 426-430: “Furthermore, SO₂ and H₂O are typical components in most industrial flue gas, especially for non-ferrous metal smelting flue gas, which is characterized by high Hg⁰, high SO₂, and high H₂O concentrations¹⁸. The proposed self-sustained Al₂O₃@MS-S_d adsorbents are well-suited for these flue gas conditions and can take full advantage of the flue gas components, turning harmful species into beneficial ones.”

9) Lines 441-451, the references to different kinetic models should be cited.

Response: Thanks for your careful review. We have cited the related references to different kinetic models, including pseudo-first-order model, pseudo-second order model, intra-particle diffusion model, and Elovich model, in the revised manuscript. \

Revised:

Line 516-518: “Different kinetic models, including pseudo-first-order model⁵⁴, pseudo-second order model⁵⁵, intra-particle diffusion model⁵⁶, and Elovich model⁵⁷, were applied to analyze the Hg⁰ adsorption rate of as-prepared sorbents.”

References:

- 54 Azizian, S. Kinetic models of sorption: a theoretical analysis. *J. Colloid Interface Sci.* **276**, 47-52, doi:<https://doi.org/10.1016/j.jcis.2004.03.048> (2004).
- 55 Ho, Y. S. & McKay, G. Pseudo-second order model for sorption processes. *Process Biochem.* **34**, 451-465, doi:[https://doi.org/10.1016/S0032-9592\(98\)00112-5](https://doi.org/10.1016/S0032-9592(98)00112-5) (1999).
- 56 Mi, X. *et al.* Preparation of graphene oxide aerogel and its adsorption for Cu²⁺ ions. *Carbon* **50**, 4856-4864, doi:<https://doi.org/10.1016/j.carbon.2012.06.013> (2012).
- 57 Ho, Y. S. & McKay, G. A Comparison of Chemisorption Kinetic Models Applied to Pollutant Removal on Various Sorbents. *Process Safety and*

Environmental Protection **76**, 332-340,
doi:<https://doi.org/10.1205/095758298529696> (1998).

Special thanks to you for your comments!

Sincerely,

Naiqiang Yan

REVIEWER COMMENTS

Reviewer #1 (Remarks to the Author):

Compared to the original manuscript, the revised manuscript has been improved a little in quality. However, the revised manuscript is still insufficient in innovation, the analysis of HgO adsorption mechanism is not deep enough, and the lack of new research methods. Therefore, this manuscript is not recommended for publication. The specific comment as follows:

(1) The process of activating MS by SCVD method is the main of this work. In the synthesis process, the addition order of SO₂ and H₂S is different, resulting in different results, wonder that the early addition of SO₂ would have another vital role. For example, during the reaction, is it because SO₂ reacts with MS in advance to produce a new transition state or leads to more defects in MS (unsaturated coordination environments, visual characterization, metal-sulfur (M-S) bond energy corresponds to vacancy defect or energy?). As more active Sn²⁺ sites are obtained, the authors should pay more attention to the mechanism of this reaction process and need to provide more microscopic evidences (defective surface structure information) in comparison with HgO adsorption conditions. XPS, XRF, Roman and HgO-TPD are not in-situ technology, I suggest authors should give new views, differing from the common descriptions to attract readers' eyes.

(2) The adsorption mechanism of HgO and metal sulfide has been reported in many similar literatures, the length of this part can be shortened appropriately.

(3) If HgO is desorbed after adsorption by SCVD-MS, will MS recover its original activity? By convention, this will free up more space for the next SCVD. And SCVD process requires both SO₂ and H₂S, in the actual industrial flue gas, whether there are such conditions to meet this condition?

(4) HgS can be used as a new material to activate Sd because Hg atom has similar properties to metals such as Cu according to the description given by the author. Will CuS nanoparticles (Provided that they are not treated with SCVD) exhibit the same performance as SCVD-CuS after the adsorption of HgO and then treated with SCVD? The authors also pointed out that SO₂ will also lead to the emergence of more active S during the adsorption of HgO, which is a promotion effect on adsorption. Then, does SO₂ have a similar effect during the adsorption of HgO by MS without SCVD?

Reviewer #2 (Remarks to the Author):

The authors had well addressed all my comments. It can be accepted in current form.

Reviewer #3 (Remarks to the Author):

Mercury pollution control is a global issue. This study innovatively put forward a in-situ low-temperature

sulfur CVD strategy to capture elemental mercury from flue gas. All questions have been answered. I advice to accept this manscript.

REVIEWER COMMENTS

Reviewer #1 (Remarks to the Author):

Compared to the original manuscript, the revised manuscript has been improved a little in quality. However, the revised manuscript is still insufficient in innovation, the analysis of Hg^0 adsorption mechanism is not deep enough, and the lack of new research methods. Therefore, this manuscript is not recommended for publication. The specific comment as follows:

(1) The process of activating MS by S-CVD method is the main of this work. In the synthesis process, the addition order of SO_2 and H_2S is different, resulting in different results, wonder that the early addition of SO_2 would have another vital role.

i) For example, during the reaction, is it because SO_2 reacts with MS in advance to produce transition state or leads to more defects in MS (unsaturated coordination environments, visual characterization, metal-sulfur (M-S) bond energy corresponds to vacancy defect or energy)?

Response: Thanks for your comment. Yes, the addition order of SO_2 and H_2S is an important factor that would affect the performances, and we had conducted the related experiments in the original manuscript (Supplementary Fig. 3). According to your advice, we re-checked such effects carefully. The results demonstrated that the activity of $\text{Al}_2\text{O}_3@\text{ZnS}$ pretreated with H_2S followed by SO_2 , was significantly lower than that of $\text{Al}_2\text{O}_3@\text{ZnS}$ pretreated with SO_2 followed by H_2S , which indicates the Eley-Rideal mechanism of S^0 deposition.

Supplementary Fig. 3 Effect of the addition sequence of H₂S and SO₂ on the Hg⁰ adsorption performance of Al₂O₃@ZnS-S₄ during S-CVD process. Reaction conditions: sorbent mass = 0.3 g, total flow rate = 360 mL min⁻¹, S-CVD time = 15 min, Hg⁰ concentration = (1.5±0.05) mg m⁻³, reaction temperature = 80 °C.

For comparison, we further supplementary the effect of single SO₂ pretreatment on the Hg⁰ adsorption performance, which is similar to that of raw Al₂O₃@ZnS. The results of Raman and XPS spectra also excluded the possibility of the produce of transition state or defects on Al₂O₃@ZnS.

Fig. R1 a Raman spectra and **b** XPS spectra of Al₂O₃@ZnS, SO₂ treated Al₂O₃@ZnS, and SO₂+H₂S treated Al₂O₃@ZnS (Al₂O₃@ZnS-S₄).

Therefore, the conclusion can be made that the pretreatment of SO₂ to MS is slightly helpful to the follow-up of H₂S for S-CVD, but it is hard to promote Hg⁰ adsorption only by itself. The pretreatment of SO₂ to MS cannot produce any useful defects for Hg⁰ adsorption, but it may bring out any new defects to enhance the following S-CVD process.

Revised:

Line 93-94: “While, pretreatment only by SO₂ cannot enhance the activity of Al₂O₃@MS.”

ii) As more active S_n^{2-} sites are obtained, the authors should pay more attention to the mechanism of this reaction process and need to provide more microscopic evidences (defective surface structure information) in comparison with Hg° adsorption conditions. XPS, XRF, Raman and Hg° -TPD are not in-situ technology, I suggest authors should give new views, differing from the common descriptions to attract readers' eyes.

Response: Thanks for your suggestion. In the original manuscript, the existing characterization results, including XRD, XPS, XRF, Raman and Hg° -TPD, have been able to demonstrate the generation of S_n^{2-} . As you know, sulfur in S-CVD is slightly volatile and it readily deposit on the optical lenses to make the signal attenuated or even collapsed, which make such in-situ tests more difficult or unapplicable. However, we still tried a lot on the in-situ techniques or even the sulfur-species frozen method with propylene (C_3H_6) has ever been employed (please see as follows).

To gain insight into the formation mechanism of S_n^{2-} on $Al_2O_3@ZnS-S_d$, pure ZnS was further synthesized in the same way without adding Al_2O_3 pellets to directly serve as support for S-CVD process. The microscopic images indicate the deposition of S_d° on ZnS surface resulted in the formation of Zn defects (Supplementary Fig. 14), which is in line with the results of XPS spectra that demonstrated the generation of unsaturated coordinated S_n^{2-} after S_d° deposition of $Al_2O_3@ZnS$.

Supplementary Fig. 14 STEM-HAADF images of **a** ZnS and **b** ZnS-S_d. **c** and **d** The corresponding simulated elemental map of zinc atoms marked in a and b.

Further, in-situ heating Raman spectroscopy method was employed to investigate the

dynamic evolution of sulfur species on ZnS-S_d (Supplementary Fig. 15). The results reveals that the elevated temperature led to the ring-opening of S₈ and the formation of S_n²⁻ chain.

Supplementary Fig. 15 Raman spectra of ZnS and in-situ Raman spectra of ZnS-S_d at temperature range of 25–140 °C.

Besides, to demonstrate the formation of S_n²⁻ chains on Al₂O₃@ZnS-S_d, C₃H₆ as a cross-linker was used to froze the sulfur species. The adsorption amount over Al₂O₃@ZnS-S_d was calculated as 0.16 and 1.30 mmol at 80 and 120 °C (Supplementary Fig. 16a), indicating an increase in S_n²⁻ content. Moreover, the Fourier transform infrared spectroscopy (FTIR) and ¹³C nuclear magnetic resonance (NMR) spectra demonstrated the formation of C–S bonds (Supplementary Fig. 16b and c); while, no peaks related to C=C were found. Thus, this further verifies the existence of S_n²⁻ chain on Al₂O₃@ZnS-S_d.

Supplementary Fig. 16 a C₃H₆ adsorption breakthrough curves of Al₂O₃@ZnS-S_d at 80 and 120 °C. Reaction conditions: sorbent weight = 0.3 g, flow rate = 500 mL min⁻¹,

C₃H₆ concentration = 4000 ppm. **b** Fourier transform infrared spectroscopy (FTIR) spectra of Al₂O₃@ZnS, Al₂O₃@ZnS-S_d and Al₂O₃@ZnS-S_d after adsorption of C₃H₆ at 80 and 120 °C. **c** ¹³C nuclear magnetic resonance (NMR) spectra of Al₂O₃@ZnS-S_d after adsorption of C₃H₆ at 80, 100 and 120 °C.

Based on the additional experiments results and the new insights into the S_d⁰ activation mechanism, the related contents have been revised as follows:

Revised:

Line 248-256: “To better observe the microscopic changes and the dynamic evolution of deposited S_d⁰, which is also demonstrated by Fourier transform infrared spectroscopy (FTIR) and ¹³C nuclear magnetic resonance (NMR) using propylene as an indicator (Supplementary Fig. 16)”

(2) The adsorption mechanism of Hg⁰ and metal sulfide has been reported in many similar literatures, the length of this part can be shortened appropriately.

Response: Thanks for your comment. We have shorten the section, and just focused on the deposition and activation mechanism of S_d⁰ and self-sustained adsorption mechanism of Hg⁰, which is quite different from literatures.

We have changed the subheading of “Hg⁰ adsorption mechanism over Al₂O₃@MS-S_d” to “S_d⁰ activation and Hg⁰ adsorption mechanisms over Al₂O₃@MS-S_d” and devoted a large portion to analyzing and discussing the activation mechanism of S_d⁰ over Al₂O₃@MS-S_d.

Revised:

Line 225: “Further, to investigate the S_d⁰ activation mechanism, ...”

Line 252-256: “Furthermore, in-situ Raman spectra revealed that with temperature increased from 25 to 120 °C, the vibration modes of S₈ in ZnS-S_d gradually converted to S_n²⁻ (Supplementary Fig. 15), which is also demonstrated by Fourier transform infrared spectroscopy (FTIR) and ¹³C nuclear magnetic resonance (NMR) using propylene as an indicator (Supplementary Fig. 16)”

(3) i) If Hg⁰ is desorbed after adsorption by S-CVD-MS, will MS recover its original

activity? By convention, this will free up more space for the next S-CVD.

Response: Thanks for your comment. This work proposed the S-CVD strategy to realize the self-sustained adsorption of Hg^0 , which, as mentioned in the Introduction, enables the in-situ reactivation without replacing the spent adsorbents. If Hg^0 is to be desorbed after each adsorption on $\text{Al}_2\text{O}_3@\text{MS-S}_a$, the used adsorbents would need to be treated at high temperatures offline and replaced with new adsorbents.

Despite that, we further supplemented experiments to demonstrate whether $\text{Al}_2\text{O}_3@\text{MS}$ can regain activity after Hg^0 desorption. As shown in Supplementary Fig. 19, after Hg^0 desorption, the spent $\text{Al}_2\text{O}_3@\text{ZnS-S}_a$ almost lost its activity. As anticipated, after second round of S-CVD, $\text{Al}_2\text{O}_3@\text{ZnS-S}_a$ regained its original activity.

Supplementary Fig. 19 Effect of the Hg^0 adsorption-desorption cycle on the performance of $\text{Al}_2\text{O}_3@\text{ZnS-S}_a$. Reaction conditions: adsorbent weight = 0.4 g, adsorption temperature = 120 °C, desorption temperature = 50–550 °C (heating rate = 5 °C min^{-1}), S-CVD time = 30 min (5000 ppm SO_2 + 100 ppm H_2S), total flow rate = 360 mL min^{-1} .

Revised:

Line 293-294: “Moreover, $\text{Al}_2\text{O}_3@\text{ZnS-S}_a$ can restore its original activity after Hg^0 desorption and secondary S-CVD (Supplementary Fig. 19).”

ii) And S-CVD process requires both SO_2 and H_2S , in the actual industrial flue gas, whether there are such conditions to meet this condition?

Response: Thanks for your comment. As is known, SO_2 is widely present in industrial flue gases with mercury, e.g., coal-fired flue gases with 0.02%-0.5% of

SO₂ and 0.1%-10% for most non-ferrous smelting flue gases. Meanwhile, H₂S or its raw materials (Na₂S or NaHS) are readily accessible as they are often used for heavy metals removal from various industrial wastewaters; moreover, methods, including anaerobic membrane bioreactor, CS₂ hydrolysis, and directly reduction from SO₂, etc., have been developed for H₂S production.

We have added in the section of Introduction a description of the sources of H₂S. The related contents are shown as follows:

Revised:

Line 56-58: “Fortunately, H₂S or its raw materials (Na₂S or NaHS) are easily accessible, commonly used for heavy metals removal from various wastewaters in non-ferrous smelters 22,23.”

(4) HgS can be used as a new material to activate S_d because Hg atom has similar properties to metals such as Cu according to the description given by the author.

i) Will CuS nanoparticles (Provided that they are not treated with S-CVD) exhibit the same performance as S-CVD-CuS after the adsorption of Hg⁰ and then treated with S-CVD?

Response: Thanks for your comment. **Yes, CuS also shows the similar properties to ZnS and is even more effective at low temperatures (Fig. 1 b and Supplementary 25). Because sulfur species on CuS are more complex than those on ZnS, it is difficult to identify what are from CuS itself or S-CVD at the beginning pahse. Therefore, we select ZnS as the model of MS to clarify the activation mechanism of S-CVD on MS.**

Fig. 1. b Hg⁰ adsorption capacities of different Al₂O₃@MS and Al₂O₃@MS-S_d (M = Cu, Zn, Cd, In, Pb, Sn, Ni, Co, Fe, and Mn).

As shown in Fig. 1 b, the Hg^0 adsorption capacity of $\text{Al}_2\text{O}_3@\text{CuS-S}_d$ is far higher than that of $\text{Al}_2\text{O}_3@\text{CuS}$. Moreover, as shown in Fig. R2, the Hg^0 adsorption efficiency of $\text{Al}_2\text{O}_3@\text{CuS}$ decreased to ~20% within ~5 h; while 30 min of in-situ S-CVD increased its Hg^0 removal efficiency to ~97% and only decreased to ~88% after ~5 h.

Fig. R2 Hg^0 adsorption curve of $\text{Al}_2\text{O}_3@\text{CuS}$ and $\text{Al}_2\text{O}_3@\text{CuS-S}_d$ (after depletion of initial activity). Reaction conditions: sorbent mass = 0.4 g, temperature = 60 °C, $[\text{Hg}^0] = (2.5 \pm 0.05) \text{ mg m}^{-3}$, $[\text{SO}_2] = 5000 \text{ ppm}$, $[\text{H}_2\text{O}] = 4\%$, S-CVD time = 30 min (100 ppm H_2S), and total flow rate = 300 mL min^{-1} .

ii) The authors also pointed out that SO_2 will also lead to the emergence of more active S during the adsorption of Hg^0 , which is a promotion effect on adsorption. Then, does SO_2 have a similar effect during the adsorption of Hg^0 by MS without S-CVD?

Response: Thanks for your comment. **The query is almost the same as the first one, and we have explained in detail there. Additional experiments and explanations are as follows:**

As shown in Supplementary Fig. 12, the addition of 1% SO_2 barely has effect on the performance of $\text{Al}_2\text{O}_3@\text{CuS}$, while a further increase of SO_2 concentration to 4% and 6% resulted in the decrease of Hg^0 removal efficiency to 79% and 52%, respectively, at 1440 min. Therefore, without the assistance of S-CVD, SO_2 is unable to exert its positive effect on the Hg^0 adsorption over MS. This is due to the relatively stable and compact structure of MS, making it difficult for the surface S^{2-} sites to interact with SO_2 and form more active sulfur sites.

Supplementary Fig. 11 The effect of SO₂ concentration on the Hg⁰ adsorption performance of Al₂O₃@CuS without S-C\TD. Reaction conditions: sorbent mass = 5 g, temperature = 80 °C, flow rate = 200 mL min⁻¹, and [Hg⁰]_{in} = 2400–2500 jig m⁻³.

Revised:

Line 204-207: “Additionally, the performance of Al₂O₃@MS without S-C\TD assistance was significantly reduced at high concentrations of SO₂, excluding the promotional effect of SO₂ on Al₂O₃@MS itself (Supplementary Fig. 11).”

Reviewer #2 (Remarks to the Author):

The authors had well addressed all my comments. It can be accepted in current form.

Response: Thanks for your positive comment!

Reviewer #3 (Remarks to the Author):

Mercury pollution control is a global issue. This study innovatively put forward a in-situ low-temperature sulfur C\TD strategy to capture elemental mercury from flue gas.

All questions have been answered. I advice to accept this manuscript.

Response: Thanks for your positive comment!

Special thanks to you for your comments!

Sincerely,
Naiqiang Yan

REVIEWER COMMENTS

Reviewer #1 (Remarks to the Author):

The revised manuscript still lacks some highlights to some extent, and the mechanism is not in-depth. Considering the current high impact, I do not recommend its publication.

Point-by-point Response to Reviewers' Comments

Dear reviewers:

Thank you for your comments concerning our manuscript entitled “In-situ Low-temperature Sulfur CVD on Metal Sulfides with SO₂ to Realize Self-sustained Adsorption of Mercury” (ID: NCOMMS-23-40042C). These comments are all valuable and helpful for further revising and improving our manuscript. We have carefully studied these comments and made the corrections. Besides, we have reorganized the manuscript to enhance the clarity and logical flow of our work. The revised portion is marked in blue in the revised manuscript. The detailed responses are as follows:

REVIEWER COMMENTS

Reviewer #1 (Remarks to the Author):

The revised manuscript still lacks some highlights to some extent, and the mechanism is not in-depth. Considering the current high impact, I do not recommend its publication.

Response: Thanks for the comments and the detailed suggestion in the attached file. We have further supplemented the XAFS, XPS depth profiling, DFT calculations to demonstrate the unsaturated coordination environments of deposited S_d on MS-S_d and validate our proposed self-sustained adsorption mechanism.

The S L-edge X-ray absorption near edge structure (XANES) pattern of ZnS-S_d showed the formation of S_n²⁻ species between S²⁻ and SO₃²⁻ (Supplementary Fig. 22a). The Zn K-edge extended X-ray absorption fine structure (EXAFS) patterns verified the presence of Zn-S bonds in ZnS and ZnS-S_d (Supplementary Fig. 22b). Moreover, the coordination number of Zn atoms in ZnS and ZnS-S_d were respectively calculated as 3.9 and 3.5 according to the EXAFS R space fitting results, confirming the formation of local unsaturated coordination environments in ZnS-S_d (Supplementary Fig. 22c). The XPS spectra of ZnS-S_d after Ar⁺ etching for different depth revealed the shift of S_n²⁻ species to lower binding energy (Supplementary Fig. 23). 2 nm of etching depth showed no significant variation, which after 4 nm and 6 nm of etching depth, the binding energy of S_n²⁻ 2p_{3/2} decreased by 0.2 eV and 0.3 eV, respectively. This confirms

that the deposition of S_d on ZnS, and the closer to the deposition interface, S_d is more capable to bond with ZnS surface atoms, thereby lowering the average valence state of S_n^{2-} , which is consistent with the decrease in Zn atom's coordination number based on the XAFS results.

Supplementary Fig. 22 **a** S L-edge XANES patterns of ZnS and ZnS- S_d . **b** Zn K-edge EXAFS patterns of ZnS, ZnS- S_d , Zn foil, and ZnO. **c** Scheme illustration of the coordination number change from ZnS to ZnS- S_d .

Supplementary Fig. 23 The XPS spectra of ZnS- S_d at different Ar^+ etching depth.

Supplementary Table 6 Zn K-edge EXAFS fitting parameters of Zn foil, ZnO, ZnS, and ZnS-S_d.

Sample	Shell	CN ^a	R (Å) ^b	σ^2 ($10^{-2} \times \text{Å}^2$) ^c	ΔE_0 (eV) ^d	R factor (Å)
Zn foil	Zn–Zn	6	2.66	1.24	2.2	0.0045
	Zn–Zn	6	2.84	3.07	4.5	
ZnO	Zn–O	4.0±0.5	1.96	0.36	3.6	0.0162
	Zn–Zn	12.1±2.1	3.24	1.54	3.9	
ZnS	Zn–S	3.9±0.2	2.33	0.61	3.6	0.0048
ZnS-S _d	Zn–S	3.5±0.3	2.43	0.58	3.1	0.0108

^aCN, coordination number; ^bR, the distance to the neighboring atom; ^c σ^2 , the mean square relative displacement (MSRD); ^d ΔE_0 , inner potential correction; R factor indicates the goodness of the fit. $S_0^2 = 0.865$ (according to the experimental EXAFS fit of Zn foil by fixing CN = 6 as the known crystallographic value. Fitting range: $3.0 \leq k$ (Å^{-1}) ≤ 10.0 and $1.0 \leq R$ (Å) ≤ 3.0 (Zn foil); $2.5 \leq k$ (Å^{-1}) ≤ 12.0 and $1.0 \leq R$ (Å) ≤ 3.2 (ZnO); $3.0 \leq k$ (Å^{-1}) ≤ 12.0 and $1.0 \leq R$ (Å) ≤ 2.3 (1-Zn); $3.0 \leq k$ (Å^{-1}) ≤ 12.0 and $1.0 \leq R$ (Å) ≤ 2.3 (2-Zn). A reasonable range of EXAFS fitting parameters: $0.700 < S_0^2 < 1.000$; $CN > 0$; $\sigma^2 > 0 \text{ Å}^2$; $|\Delta E_0| < 15 \text{ eV}$; $R \text{ factor} < 0.02 \text{ Å}$.

DFT calculations were further conducted to validate the S_d activation and Hg⁰ self-sustained adsorption mechanisms. ZnS(111) model was first established and optimized to investigate the S_d⁰ activation and Hg⁰ adsorption energy. The results demonstrated that ZnS-S₄ was the most stable structure with a negative Gibbs free energy (ΔG) of -77.9 kJ/mol from ZnS-S₈ to ZnS-S₄. Moreover, compared with ZnS, the Zn–S bond length in ZnS-S₄ surface increased from 2.31 Å to $2.38\text{--}2.48 \text{ Å}$. The ZnS-S₄ also had the highest Hg⁰ adsorption energy ($E_{\text{ads}} = -125 \text{ kJ/mol}$) than ZnS (-46.2 kJ/mol) and ZnS-S₈ (-67.6 kJ/mol). Further, ZnS@HgS structure was constructed to verify the role of HgS in S_d⁰ activation and Hg⁰ adsorption. The calculation results also proved the stable structure of ZnS@HgS-S₄ with a negative ΔG of -68.4 kJ/mol and the highest E_{ads} of -73.1 kJ/mol , confirming the proposed Hg⁰ self-sustained adsorption mechanism.

Supplementary Fig. 31 Gibbs free energy of S₈ (S_d⁰) activation on ZnS(111) surface.

Supplementary Fig. 32 Zn-S bond length in ZnS and ZnS-S₄ surface.

Fig. 5. i DFT calculations of S_d activation and Hg⁰ self-sustained adsorption behaviors on ZnS(111) surface.

Furthermore, to enhance the clarity and logical flow of our work, we have reorganized the manuscript. Specifically, we have provided a more explicit schematic illustration of the self-sustained adsorption of Hg^0 on metal sulfides (Fig. 1). Notably, the section on “ S_d^0 activation and Hg^0 adsorption mechanisms over $\text{Al}_2\text{O}_3@\text{MS-S}_d$ ” and “Self-sustained Hg^0 adsorption mechanism over $\text{Al}_2\text{O}_3@\text{MS-S}_d$ ” from the previous version of manuscript have been consolidated into the section titled “Mechanism for self-sustained Hg^0 adsorption on $\text{Al}_2\text{O}_3@\text{MS-S}_d$ ”.

Fig. 1. a Hg^0 removal through proposed in-situ S-CVD strategy in smelting flue gas. **b** Schematic illustration of the Hg^0 self-sustained adsorption on metal sulfides. **b1** Initial stage, S_d^0 activated only by MS; **b2** Transition stage, S_d^0 activated by MS and/or HgS ; **b3** Sustained stage, S_d^0 activated by HgS itself.

Besides, to the best of our knowledge, the S-CVD strategy for self-sustained Hg^0 adsorption proposed for the first time, breaking the limitations of adsorption capacity of traditional materials and representing a breakthrough shift in adsorption modes.

Meanwhile, this research is also innovative in the following aspects:

1) Unlike traditional disposable adsorbents, the S-CVD strategy can continuously replenish the active sites on MS after the saturation of Hg^0 adsorption, thereby realizing self-sustained adsorption.

2) Any natural sulfide ores containing chalcophile metal elements (e.g., chalcopyrite and sphalerite) can also directly serve as adsorbents for self-sustained adsorption of Hg^0 through S-CVD strategy, significantly reducing the cost of Hg^0 treatment.

3) In addition to Hg^0 , this S-CVD strategy holds potential applications for other SO_2 -containing scenarios requiring MS in heavy metal removal.

Revised:

Line 101-110: “In actual non-ferrous smelting processes, flue gas particle-bound mercury (Hg^p) and oxidized mercury (Hg^{2+}) can be respectively removed by an electrical precipitator and scrubber, resulting in a subsequent flue gas with high concentrations of SO_2 and Hg^0 (Supplementary Fig. 2). Extraction of approximately 0.1% of total SO_2 for on-site conversion to H_2S ^{33,34} can satisfy the needs of S-CVD, and Hg^0 will be removed by the proposed self-sustained adsorption method on adsorbents (Fig. 1a). In this method, MS functions as the fresh surface for the initial S-CVD and Hg^0 adsorption, and then the spent MS (i.e., MS-HgS) acts as new surface for further S-CVD and Hg^0 adsorption, ultimately achieving the sustained adsorption by HgS itself (Fig. 1b).”

Line 336-343: “The XAFS S L-edge spectra confirmed the formation of S_n^{2-} species in ZnS- S_d (Supplementary Fig. 22a). The extended XAFS (EXAFS) Zn K-edge spectra further revealed a decrease from 3.9 to 3.5 in the coordination number of Zn to S atoms in ZnS- S_d compared to that in pristine ZnS (Supplementary Fig. 22b and c, and Supplementary Table 6), further certifying the formation of unsaturated coordination sites. The XPS depth profiling results depicted that with the Ar^+ etching depth increased to 6 mm, the average valence state of S_n^{2-} decreased, suggesting a shortening of the S_n^{2-} chain length close to the ZnS surface (Supplementary Fig. 23).”

Line 397-416: “DFT calculations were applied to elaborate the Hg^0 self-sustained

adsorption mechanism on ZnS. ZnS (111) model was established and optimized to investigate the Gibbs free energy (ΔG) of S_d^0 activation and the adsorption energy (E_{ads}) of Hg^0 . Given that S_n^{2-} dominated in the Hg^0 adsorption process, the ΔG from S_8 ring to S_n ($n = 6, 4, 2,$ and 1) chains were first calculated. As presents in Supplementary Fig. 31, the ZnS- S_8 can spontaneously convert to ZnS- S_6 and then to ZnS- S_4 with a negative ΔG of -77.9 kJ/mol, while the conversion of ZnS- S_4 to ZnS- S_2 has a positive ΔG of 158.6 kJ/mol. This indicates the most stable structure of ZnS- S_4 . Moreover, compared with ZnS, the Zn-S bond length in ZnS- S_4 surface increased from 2.31 Å to 2.38 – 2.48 Å (Supplementary Fig. 32), in lined with the decrease in Zn-S coordination number in ZnS- S_d . The Hg^0 adsorption on ZnS, ZnS- S_8 , and ZnS- S_4 were further optimized and calculated as -46.2 , -67.6 , and -125.0 kJ/mol, respectively (Fig. 5i). The highest E_{ads} of ZnS- S_4 verifies the important role of S_4 chain on Hg^0 adsorption. Further, considering the formation of HgS on $Al_2O_3@ZnS-S_d$ after Hg^0 self-sustained adsorption, we constructed the structure of ZnS@HgS for subsequent S_d^0 activation and Hg^0 adsorption. The negative ΔG (-68.4 kJ/mol) from ZnS@HgS- S_8 to ZnS@HgS- S_4 demonstrated its spontaneous conversion process. The E_{ads} of Hg^0 on ZnS@HgS- S_4 (-73.1 kJ/mol) was higher than those of ZnS@HgS (26.0 kJ/mol) and ZnS@HgS- S_8 (-27.9 kJ/mol). This confirms the role of HgS in the activation of S_d^0 and further adsorption of Hg^0 , supporting the proposed Hg^0 self-sustained adsorption mechanism.”

Special thanks to you for your comments!

Sincerely,
Naiqiang Yan

REVIEWERS' COMMENTS

Reviewer #1 (Remarks to the Author):

Although the authors add some characterization results, I still do not see some new ideas about mercury adsorption on metal sulfide surfaces. Therefore, I do not recommend publication.